# Sigma models on flags

**Kantaro Ohmori, Nathan Seiberg and Shu-Heng Shao**

School of Natural Sciences, Institute for Advanced Study, Princeton, NJ 08540, USA

## Abstract

We study (1+1)-dimensional non-linear sigma models whose target space is the flag manifold $\frac{U(N)}{U(N_1) \times U(N_2) \cdots U(N_m)}$, with a specific focus on the special case $U(N)/U(1)^N$. These generalize the well-known $\mathbb{CP}^{N-1}$ model. The general flag model exhibits several new elements that are not present in the special case of the $\mathbb{CP}^{N-1}$ model. It depends on more parameters, its global symmetry can be larger, and its 't Hooft anomalies can be more subtle. Our discussion based on symmetry and anomaly suggests that for certain choices of the integers $N_I$ and for specific values of the parameters the model is gapless in the IR and is described by an $SU(N)_1$ WZW model. Some of the techniques we present can also be applied to other cases.



# 1  Introduction

The goal of this paper is to explore two-dimensional sigma model whose target space is a generalized flag manifold

$$\mathcal{M}_{N_1,N_2,\ldots,N_m} = \frac{U(N)}{U(N_1) \times U(N_2) \cdots U(N_m)}$$
$$\sum_{I=1}^{m} N_I = N \; . \tag{1.1}$$

A special familiar case is $m = 2$ where $\mathcal{M}_{n,N-n}$ is a Grassmannian and an even more special case is $\mathcal{M}_{1,N-1} = \mathbb{CP}^{N-1}$. As we will see, the generic case exhibits a number of new elements that are not present in the familiar $\mathbb{CP}^{N-1}$ sigma model.

The $\mathbb{CP}^{N-1}$ model depends on a single relevant parameter, the overall size of the target space $\sqrt{r}$; i.e. the metric is proportional to $r$ and perturbation theory is an expansion in $\frac{1}{r}$. The theory is asymptotically free, as $r$ shrinks in the IR. In addition, the theory depends on a $2\pi$-periodic $\theta$-parameter. A combination of techniques has shown that for generic $\theta$ the model is gapped and the $\theta$ dependence is smooth. The only exception is the physics at $\theta = \pi$, where the system has another $\mathbb{Z}_2$ global symmetry. For generic $N$ this $\mathbb{Z}_2$ symmetry is spontaneously

broken there and the system has two gapped vacua. For $N = 2$ the system is gapless and the IR dynamics is that of the $SU(2)_1$ Wess-Zumino-Witten (WZW) model. See, for example, the introduction of [1] and references therein for a review of this classic story.

The main difference between the $\mathbb{CP}^{N-1}$ model and its generalization $\mathcal{M}_{N_1, N_2, \ldots, N_m}$ is that the latter depends on more parameters. For example, the more general model depends on $m-1$ $2\pi$-periodic $\theta$-parameters. Other continuous parameters arise because the $SU(N)$ invariant metric on $\mathcal{M}_{N_1, N_2, \ldots, N_m}$ is not unique. Finally, we can also have $SU(N)$ invariant two-form background fields $B$. There are various loci on the parameter space where the model has enhanced discrete global symmetry, which can be imposed to constrain the renormalization group flow.

In most of the paper we will focus on the extreme case $m = N$ where the target space is the flag manifold

$$\mathcal{M} = \mathcal{M}_{1, 1, \ldots, 1} = \frac{U(N)}{U(1)^N} = \frac{SU(N)}{U(1)^{N-1}} \ . \tag{1.2}$$

The model has a global $PSU(N)$ symmetry (the center of the naive $SU(N)$ symmetry acts trivially on all the physical operators). The sigma model on $\mathcal{M}$ can be thought of as a different $SU(N)$ generalization of the $\mathbb{CP}^1$ model than the $\mathbb{CP}^{N-1}$ model. By contrast to the $\mathbb{CP}^{N-1}$ model with $N > 2$, we will argue below that the flag sigma model $\mathcal{M}$ has an interesting gapless phase.

Returning to the general $N$ case, the theory depends on $\frac{N(N-1)}{2}$ continuous $PSU(N)$ invariant metric parameters and $\frac{N(N-1)}{2}$ continuous $PSU(N)$ invariant $B$ parameters. $N-1$ of the $B$ parameters have $H = dB = 0$ and they lead to $N-1$ $\theta$-parameters. The remaining $B$ parameters label deformations with nonzero $H = dB$.

On certain subspaces of the parameter space the model has additional global symmetries. The most symmetric model has an $\mathbb{S}_N$ permutation symmetry. We will discuss the details of this symmetry below. Imposing this symmetry most of the parameters of the metric and the two-form $B$ deformations are set to zero and the theory depends only on the overall scale of the target space $r$. In the IR this model is expected to be gapped and trivial. Therefore, it is interesting to impose only a smaller discrete symmetry.

An interesting symmetry to impose, in addition to the obvious $PSU(N)$, is $\mathbb{Z}_N$. We will see that if we impose this symmetry, the model depends on $\lfloor \frac{N}{2} \rfloor$ continuous metric parameters and $\lfloor \frac{N-1}{2} \rfloor$ continuous $B$ parameters. In addition, there are some discrete $B$ parameters, which are associated with nontrivial $\theta$-parameters. Depending on the values of these discrete parameters, the model has a mixed anomaly between the $PSU(N)$ global symmetry and the discrete $\mathbb{Z}_N$ symmetry. These anomalies arise from the fact that the Lagrangian of the theory with these values of the $\theta$-parameters is not invariant under the global symmetries unless we use the $2\pi$ periodicity of the $\theta$-parameters. The existence of these anomalies means that the IR theory cannot be trivial. The global symmetry could be spontaneously broken, or the system could be gapless, or it could be gapped with some topological quantum field theory (TQFT).[1] We will argue that in some of these cases the long distance behavior of the system is gapless and it is described by the $SU(N)_1$ WZW model.

The global symmetries we have discussed so far are the global symmetries of the UV theory, $G_{UV}$. Not knowing what the IR dynamics is, we should explore various possible candidates. Since the IR symmetry $G_{IR}$ can be larger than the UV symmetry, we should examine how it could be embedded in it $G_{UV} \subset G_{IR}$. In particular, $PSU(N)$ is always a global symmetry of our UV model. In a gapless phase, the $PSU(N)$ global symmetry necessarily enhances to a full-fledged $\mathfrak{su}(N)_L \times \mathfrak{su}(N)_R$ current algebra. The most natural and minimal candidate for the

---

[1]The case of spontaneous global symmetry breaking is a special case of such a TQFT.

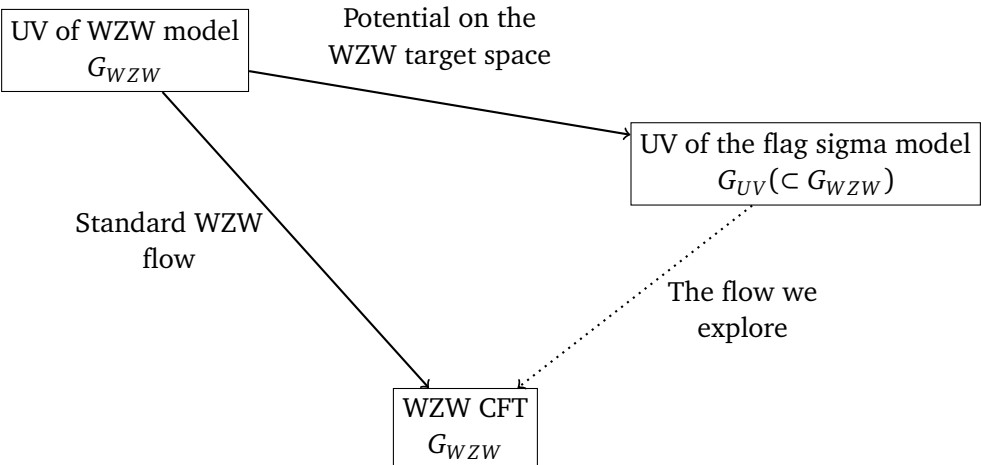

Figure 1: The UV WZW model flows in the IR to the WZW CFT. This flow preserves the full $G_{WZW}$ global symmetry. Here we deform this UV theory by a potential that restricts the field to take values in the flag $\mathcal{M}$ and breaks the WZW symmetry to $G_{UV} \subset G_{WZW}$. Then we explore whether the sigma model can flow to the WZW CFT (dashed line in the diagram). The global symmetry of each theory in the figure is written below it.

IR conformal field theory (CFT) is therefore the $SU(N)$ WZW model, which has

$$G_{IR} = G_{WZW} = \frac{SU(N)_L \times SU(N)_R}{\mathbb{Z}_N} \rtimes \mathbb{Z}_2^C \tag{1.3}$$

global symmetry, as well as parity symmetries.[2] Once we find how the symmetries are embedded, we should make sure that they have the same anomalies. We will present a powerful tool to do that.

We would like to examine whether our flag sigma model can flow to the WZW CFT. The WZW Lagrangian, which is a group manifold sigma model at large radius plus a quantized Wess-Zumino (WZ) term, gives us a flow from a free UV theory to a nontrivial conformal field theory in the IR [2]. Unlike our flag theory, here the global symmetry (1.3) and its various anomalies are present throughout the renormalization group flow (see the left flow in Figure 1). Then we will turn on a suitably chosen potential in the UV of this model that restricts the WZW field to take values in a subspace of field space. We will arrange it such that this subspace is our flag target space. This method has been used in [3,4]. This potential breaks the WZW symmetry $G_{WZW}$ to a subgroup $G_{UV} \subset G_{WZW}$, which is the UV symmetry of our flag sigma model. Clearly, when the coefficient of the new potential terms is large we first flow from the UV of the WZW to our sigma model (the top flow in Figure 1) and then we flow from there to the IR. We explore whether this last flow is to the WZW CFT (the dashed flow in Figure 1).

One aspect of this construction is that it guarantees that our UV symmetry $G_{UV}$ is properly embedded in the symmetry $G_{WZW}$ of the explored IR behavior and that they have the same anomalies.

Another aspect of this construction is that it gives us another tool to examine such a possible flow. Consider the WZW conformal field theory. If the dashed flow in Figure 1 exists, then it reaches the WZW point along an irrelevant operator, which is invariant under $G_{UV}$ but not invariant under $G_{WZW}$. Furthermore, we would like all the $G_{UV}$-invariant deformations of the WZW theory that are not $G_{WZW}$ invariant to be irrelevant. This would guarantee that if the flow (the dashed line in Figure 1) arrives close to the WZW point, it will be attracted to

---

[2]$\mathbb{Z}_2^C$ is charge conjugation and it is absent for $N = 2$. In Section 7 we will discuss the action of these symmetries in more detail.

it. Alternatively, if the WZW model has a relevant $G_{UV}$-invariant deformation (other than the identity operator), then a generic flow from the IR would miss it. See, for example, [5,6] for applications of this type of argument.

Specifically, we will study the theory with a global $G_{UV} = (PSU(N) \times \mathbb{Z}_N) \rtimes \mathbb{Z}_2^C$ symmetry and non-trivial $\theta$-angles, generalizing the $N = 2$ case of $\mathbb{CP}^1$ model at $\theta = \pi$.[3] We will argue that it flows to the $SU(N)_1$ WZW model where the global symmetry is enhanced to $G_{WZW}$. First, we will show that we can add a potential to the WZW model to restrict the target space to be the flag manifold. This guarantees that the these two models have the same anomaly under $G_{UV}$. Second, it is significant that the only $G_{UV}$-invariant relevant operator in the WZW CFT is the identity operator. This means that for a range of parameters the flow from the sigma model can hit this fixed point. In other words, no fine tuning is necessary.

The $SU(N)/U(1)^{N-1}$ sigma model was derived from the spin chain in [7,8]. The $N = 3$ case was subsequently studied in detail in [1]. The authors proposed that the $\mathbb{Z}_3$-symmetric model with nontrivial $\theta$-angles flows to the $SU(3)_1$ WZW model. The global symmetry and anomaly of the flag manifold for general $N$ were later studied in [4]. Their proposal about the IR phases is similar, but not identical, to ours.

What if we start with the $\mathbb{S}_N$ invariant model with trivial $\theta$-angles? In this case the UV global symmetry is

$$G'_{UV} = (PSU(N) \times \mathbb{S}_N) \rtimes \mathbb{Z}_2^C \, . \tag{1.4}$$

In the $N = 2$ case this reduces to the $\mathbb{CP}^1$ model at $\theta = 0$.

Can this $\mathbb{S}_N$ invariant model flow to the $SU(N)_1$ WZW CFT? If so, the symmetry $G'_{UV}$ cannot be embedded into $G_{WZW}$ as in the previous case. One possibility is that the $\mathbb{S}_N \subset G'_{UV}$ is unbroken and acts trivially in the IR, and there is another emergent $\mathbb{Z}_N$ symmetry that combines with $PSU(N)$ to form $G_{WZW}$. In this case we can no longer use the $\mathbb{Z}_N \subset \mathbb{S}_N$ to restrict the relevant deformations in the $SU(N)_1$ WZW model. In fact, there are many $PSU(N)$ invariant relevant deformations in the $SU(N)_1$ WZW model. For the higher level WZW models there are even more symmetry-preserving relevant deformations. Hence, we do not expect the model to hit the WZW fixed point without fine tuning. An identical argument carries over as long as the flag sigma model on $\mathcal{M}$ does not have the global symmetry $PSU(N) \times \mathbb{Z}_N$ with the same anomaly as in the $SU(N)_1$ WZW model. Another possibility is that the entire $G'_{UV}$ symmetry decouples in the IR.[4] In this case we do not expect the IR phase to be gapless because there is no symmetry argument to forbid any relevant deformation in the candidate CFT.[5]

An important part of analyzing the renormalization group flow from the flag sigma model is to explore its various operators in the UV. Here we should turn on all possible operators that are invariant under $G_{UV}$ and explore their beta-function. We will not do it in full generality. Instead, we will expand around the $\mathbb{S}_N$ invariant theory with only one parameter, the overall size square $r$. This theory has $G'_{UV} = (PSU(N) \times \mathbb{S}_N) \rtimes \mathbb{Z}_2^C$ global symmetry. We will explore small $\mathbb{Z}_N$ invariant deformations of the metric and the background $B$ around this model. At leading order this will allow us to organize them in terms of $\mathbb{S}_N$ representations. We will examine the renormalization group flow for this range of parameters and will see that for $N > 6$ the $\mathbb{S}_N$ violating but $\mathbb{Z}_N$ invariant deformations are irrelevant.[6]

---

[3]Similar to the WZW model, $\mathbb{Z}_2^C$ is absent in the $\mathbb{CP}^1$ model compared to the higher $SU(N)/U(1)^{N-1}$.

[4]The continuous global symmetry $PSU(N)$ cannot be spontaneously broken in two dimensions.

[5]Logically it is possible that the IR phase is a CFT without any relevant deformation, e.g. the $E_8$ WZW model at level 1. However, by modular invariance, it is known that any $2d$ unitary CFT with $c < 8$ must admit relevant deformations [9] (based on earlier work [10,11]). Therefore as long as the UV central charge $c_{UV} = N^2 - N$ is smaller than 8 (which is the case for $N = 2, 3$), we can confidently conclude that the IR phase must be gapped if the entire UV symmetry acts trivially in the IR. We assume the same conclusion is true for all $N$.

[6]Here, when we say irrelevant, relevant, or marginal we mean relative to the flow of the overall radius square $r$. We will explain it in detail in Section 5.

An extreme version of this range of parameters is obtained when we start with the $\mathbb{S}_N$ invariant model with its unique parameter $r$ and turn on only the $\mathbb{Z}_N$ invariant $\theta$-parameters, but not the other $\mathbb{Z}_N$ invariant deformations. Strictly, this is an unnatural thing to do. However, in perturbation theory such a setup is actually natural. Since the violation of the global $\mathbb{Z}_N$ is only due to instantons and their effect is invisible in perturbation theory, the remaining $\mathbb{S}_N$ violating but $\mathbb{Z}_N$ invariant parameters are not activated. Of course, non-perturbatively, instantons make them non-zero. Yet, it is technically natural to explore this range of parameters. The reason this range of parameters is significant is that to all orders in perturbation theory the flow preserves the global $\mathbb{S}_N$ symmetry and possible operators that can take us afar are not present.

Our general arguments fall into two classes. Kinematical considerations involve the global symmetry $G_{UV}$ in the UV and its 't Hooft anomalies. These are matched with putative IR CFTs, specifically WZW models. Once a flow from the UV sigma model to the putative IR theory is kinematically possible, we apply more dynamical considerations. These are associated with $G_{UV}$-invariant, relevant deformations of the putative IR theory. For every such operator one fine tuning is needed in order to hit the IR theory. And if no such relevant operator exists, the fixed point is attractive and no fine tuning is necessary.

However, even if no $G_{UV}$-invariant, relevant operators exist in that theory, we are still not guaranteed that the renormalization group flow from the UV indeed hits that theory. Instead, what this shows is that if the flow gets to the vicinity of that theory, it will be attracted to it. But there is no proof that the UV theory gets to the vicinity of that IR point in the first place. For this reason, whenever we say that we can hit a certain IR CFT, what we really mean is that this IR behavior is kinematically possible and if we get close to it, we end up there; i.e. that point is attractive. This falls short of a proof that the long distance behavior is indeed described by this theory.

In all our examples $G_{UV}$ has nontrivial 't Hooft anomalies, which can be matched by the proposed IR CFT. If however, as we have just said, the long distance theory is gapped, then these anomalies mean that $G_{UV}$ should be spontaneously broken and the vacuum is not unique.

The paper is organized as follows. In Section 2 we write down the Lagrangian for the $SU(N)/U(1)^{N-1}$ flag sigma model and classify the $PSU(N)$-invariant parameters. In Section 3 we discuss the discrete global symmetry of the model and its 't Hooft anomaly. In Section 4 we deform the UV WZW model to the flag sigma model, establishing the top line in Figure 1. This ensures that the global symmetry and anomaly are the same in the WZW CFT and the flag sigma model. We further discuss the symmetry-preserving relevant deformation in the IR WZW CFT point to determine whether fine-tuning is needed to hit the fixed point. Based on the above considerations, we argue that the flag sigma model with special parameters flows to the $SU(N)_1$ WZW CFT in the IR. In Section 5 we compute the one-loop beta functions of the flag sigma model and use various discrete symmetry to constrain the flow. In Section 6 we specialize to the $SU(3)/U(1)^2$ model and study the renormalization group flow in details. In Section 7 we extend our argument to the more general flag manifold. In particular we argue that the $U(rM)/U(M)^r$ sigma model with special parameters and with $r, M$ sufficiently large flows to the $SU(rM)_1$ WZW CFT. We also apply our discussion to the classic $\mathbb{CP}^{N-1}$ model and recover known results. We summarize our results in Section 8.

In Appendix A, we prove a technical identity that we use in analyzing the deformation from the UV WZW model to the flag sigma model. Appendix B counts the two-derivative deformations around the UV WZW model and matches them with the counting in the flag sigma model. In Appendix C, we use the same techniques to analyze the sigma model on the coset $SU(N)/SO(N)$. In Appendix D, we discuss some aspects of the flag sigma model with $\mathcal{N} = (2, 2)$ supersymmetry.

## 2 The Lagrangian and the Parameters

### 2.1 The Lagrangian

We are interested in a theory of several complex scalar fields $\phi_i^I$. There is an $SU(N)$ symmetry acting on the lower case index $i = 1, 2, \cdots, N$. The scalars $\phi_i^I$ are constrained to satisfy

$$\sum_i \phi_i^J \bar{\phi}_I^i = \delta_I^J \ . \tag{2.1}$$

We will also often use

$$\sum_i (\partial \phi_i^J) \bar{\phi}_I^i = -\sum_i \phi_i^J \partial \bar{\phi}_I^i \ , \tag{2.2}$$

which follows from (2.1).

We impose gauge invariance under

$$\phi_i^I \to e^{i\lambda_I} \phi_i^I \ . \tag{2.3}$$

We can do that by adding $U(1)$ gauge fields $a_I$ and use the covariant derivatives $D\phi_i^I \equiv (\partial + ia_I)\phi_i^I$ and $D\bar{\phi}_I^i \equiv (\partial - ia_I)\bar{\phi}_I^i$. We can also replace $a_I$ by

$$a_I = -\frac{i}{2} \sum_i \left( \phi_i^I \partial \bar{\phi}_I^i - \bar{\phi}_I^i \partial \phi_i^I \right) = -i \sum_i \phi_i^I \partial \bar{\phi}_I^i \ . \tag{2.4}$$

With this $a_I$ it is convenient to use

$$\sum_i \bar{\phi}_I^i D\phi_i^I = 0 \ . \tag{2.5}$$

Our model has the gauge symmetry (2.3) either in the formulation with independent gauge fields $a_I$ or if we use (2.4). Correspondingly, the scalar fields $\phi^I$ might not be single valued — we might need to cover our spacetime with patches with transition functions between them.

One special case is based on $N$ complex scalar fields $\phi_i$ and then the index $I$ is suppressed. It leads to the $\mathbb{CP}^{N-1} = \frac{U(N)}{U(1) \times U(N-1)}$ model. We will focus on the theory with $I$ ranging from 1 to $N$. The resulting theory is a nonlinear model with the target space

$$\frac{SU(N)}{U(1)^{N-1}} \ . \tag{2.6}$$

This is a different $SU(N)$ generalization of the $\mathbb{CP}^1$ sigma model.

Several comments are in order:

1. Because of (2.1), the scalars $\phi_i^I$ can be viewed as a unitary $N \times N$ matrix $\phi$ with a global $PSU(N)$ action $\phi \to V\phi$ and a local $U(1)^N$ action $\phi \to \phi M$, where $V \in SU(N)$ and $M$ a diagonal $U(N)$ matrix. However, the continuous global symmetry of the system is $PSU(N) = SU(N)/\mathbb{Z}_N$ instead of $PSU(N)$. The quotient by $\mathbb{Z}_N$ follows from the fact that each gauge invariant operator must have equal numbers of $\phi_i^I$ and $\bar{\phi}_I^j$ and therefore it transforms trivially under the center of $SU(N)$.

2. For some purposes we will find it convenient to express $\phi_i^N$ in terms of the other $\phi_i^I$. However, this might obscure some of the symmetries of the problem.

3. For $N = 2$ the two special cases $\mathbb{CP}^{N-1}$ and $SU(N)/U(1)^{N-1}$ coincide. This will allow us to compare the analysis here with the well studied $\mathbb{CP}^1$ model.

4. One important difference between the $\mathbb{CP}^{N-1}$ theory and the $SU(N)/U(1)^{N-1}$ theory (for $N > 2$) is that the $\mathbb{CP}^{N-1}$ space has a unique (up to rescaling) $SU(N)$ invariant metric, while the $SU(N)/U(1)^{N-1}$ space has a multi-parameter family of $PSU(N)$ invariant metrics. In addition, we can have a multi-parameter family of $PSU(N)$ invariant torsion terms. This means that our model is characterized by several parameters. Below we will describe these parameters and will show how they can be restricted by imposing more global symmetries.

The simplest terms in an invariant Lagrangian are

$$\mathcal{L} = \sum_{I,i} r_I |D_\mu \phi_i^I|^2 + \mathrm{i} \sum_I \frac{\theta_I}{2\pi} \epsilon^{\mu\nu} \partial_\mu a_{I\nu} , \tag{2.7}$$

where the coefficients $r_I$ and the phases $\theta_I$ are arbitrary. Here we can also substitute (2.4). Below we will impose additional discrete symmetries that will constrain the parameters $r_I$ and $\theta_I$. The Lagrangian (2.7) describes $N$ copies of the $\mathbb{CP}^{N-1}$ model that are coupled through (2.1).

What other terms can we add to (2.7)? Using (2.1) it is easy to see that the only $U(1)^N$ invariant potential without derivative is a constant. Let us move on to the two-derivative terms. $U(1)^N$ gauge invariance and the conditions (2.1) restrict the allowed terms. The general term has fields with derivatives, e.g. $D\phi D\bar\phi$, multiplied by some $\phi$'s and $\bar\phi$'s. Any $i,j$ indices that are contracted entirely within the factors without derivatives become trivial using (2.1). Therefore, we can limit ourselves to terms of the form

$$\sum_{i,j} \phi_i^I \bar\phi_J^j D\phi_j^K D\bar\phi_L^i . \tag{2.8}$$

Note that using (2.2) we can write terms like $\sum_{i,j} \bar\phi_J^j \bar\phi_L^i D\phi_j^K D\phi_i^I$ as (2.8).

We will use the formulation where the gauge field $a_I$ has been replaced by (2.4). Because of (2.5), the only two derivative term we need to consider is

$$\sum_{i,j} \phi_i^J \bar\phi_J^j (D\bar\phi_I^i)(D\phi_j^I), \qquad I \neq J . \tag{2.9}$$

Here the $a_I$'s in the covariant derivative are understood in terms of (2.4). Let us write this term as well as the original kinetic term $\sum_i |D\phi_i^I|^2$ directly in terms of $\phi$ alone:

$$\sum_i |D\phi_i^I|^2 = \sum_i |\partial \phi_i^I|^2 - |\sum_i \bar\phi_I^i \partial \phi_i^I|^2$$

$$\sum_{i,j} \phi_i^J \bar\phi_J^j (D\bar\phi_I^i)(D\phi_j^I) = \sum_{i,j} \phi_i^J \bar\phi_J^j (\partial \bar\phi_I^i - \sum_k \phi_k^I \partial \bar\phi_I^k \bar\phi_I^i)(\partial \phi_j^I + \sum_l \phi_l^I \partial \bar\phi_I^l \phi_j^I)$$

$$= (\sum_i \phi_i^J \partial \bar\phi_I^i - \sum_k \phi_k^I \partial \bar\phi_I^k \delta_I^J)(\sum_j \bar\phi_J^j \partial \phi_j^I + \sum_l \phi_l^I \partial \bar\phi_I^l \delta_J^I)$$

$$= \begin{cases} (\sum_i \phi_i^J \partial \bar\phi_I^i)(\sum_j \bar\phi_J^j \partial \phi_j^I) \text{ for } I \neq J \\ 0 \qquad \text{for } I = J . \end{cases} \tag{2.10}$$

We conclude that we can allow arbitrary $r_I$ and $\theta_I$ in (2.7) and we can have an arbitrary linear combination of term like (2.9) with real symmetric coefficients $\frac{1}{2} G_{IJ}$ (and we can set the diagonal elements $G_{II}$ to zero) with a symmetric contraction of the Lorentz indices and with arbitrary real antisymmetric coefficients $\frac{1}{2} B_{IJ}$ with an antisymmetric contraction of the Lorentz indices.

Explicitly, these two-derivative deformations terms are:

$$\sum_{1 \leq I < J \leq N} (G_{IJ}\delta^{\mu\nu} + B_{IJ}\epsilon^{\mu\nu})\left(\sum_i \phi_i^J \partial_\mu \bar{\phi}_I^i\right)\left(\sum_j \bar{\phi}_J^j \partial_\nu \phi_j^I\right). \tag{2.11}$$

The terms with $G_{IJ}$ lead to modifications of the target space metric and the terms with $B_{IJ}$ can be viewed as torsion. We can also extend the range of $I$ and $J$ and let $G_{IJ}$ be a symmetric tensor (with vanishing diagonal elements) and $B_{IJ}$ an antisymmetric tensor.

To summarize, we have the following Lagrangian for the $SU(N)$ theory parametrized by $r_I, G_{IJ}$ and $\theta_I, B_{IJ}$:

$$\sum_{I=1}^N \left(r_I\left(\sum_i |\partial \phi_i^I|^2 - |\sum_i \bar{\phi}_I^i \partial \phi_i^I|^2\right) + \frac{\theta_I}{2\pi}\epsilon^{\mu\nu}\sum_i \partial_\mu \phi_i^I \partial_\nu \bar{\phi}_I^i\right)$$
$$+ \sum_{1 \leq I < J \leq N} (G_{IJ}\delta^{\mu\nu} + B_{IJ}\epsilon^{\mu\nu})\left(\sum_i \phi_i^I \partial_\mu \bar{\phi}_J^i\right)\left(\sum_j \bar{\phi}_I^j \partial_\nu \phi_j^J\right). \tag{2.12}$$

There are redundancies between the parameters $r_I$ and $G_{IJ}$ and also between $\theta_I$ and $B_{IJ}$. To see this, let us first note that from (2.1), we have

$$\sum_I \phi_i^I \bar{\phi}_I^j = \delta_i^j, \tag{2.13}$$

from which it follows that

$$\sum_J \sum_{i,j} \phi_i^J \bar{\phi}_J^j (D\bar{\phi}_I^i)(D\phi_j^I) = \sum_i (D\bar{\phi}_I^i)(D\phi_i^I), \tag{2.14}$$

which is the same as the terms in (2.7). Therefore, we have the freedom in shifting the coefficients $r_I \rightarrow r_I + c_I$ combined with $G_{IJ} \rightarrow G_{IJ} - c_I - c_J$ (recall that $G_{IJ}$ is symmetric) and similarly, $\theta_I \rightarrow \theta_I + 2\pi b_I$ combined with $B_{IJ} \rightarrow B_{IJ} - b_I + b_J$ (recall that $B_{IJ}$ is antisymmetric). In particular, we can set $r_I = \theta_I = 0$ using this freedom. This makes it clear that the theories are labeled by $G_{IJ}$ and $B_{IJ}$. However, this might not be a convenient choice. In particular, it does not make the $2\pi$ periodicity of $\theta_I$ manifest. Alternatively, we can use this freedom to set $r_N = \theta_N = G_{IN} = B_{IN} = 0$, such that $\phi_i^N$ does not appear in the Lagrangian. This can be understood as using (2.1) to eliminate it in terms of $\phi_i^I$ with $I = 1, ..., N-1$.

A special case of the redundancy transformation is $b_I = b$ for all $I$. This shifts all the $\theta_I$ by:

$$\theta_I \rightarrow \theta_I + b, \quad \forall \; I = 1, \cdots, N, \tag{2.15}$$

without changing $B_{IJ}$. In other words, we are always free to shift all $N$ $\theta$-angles by the same amount.

With the above redundancy taken into account, we conclude that there are $N(N-1)/2$ $G$ deformation terms (including $r_I$ and $G_{IJ}$) and $N(N-1)/2$ $B$ deformation (including $\theta_I$ and $B_{IJ}$) terms preserving the $PSU(N)$ global symmetry.

## 2.2 Counterterms

The continuous global symmetry of the system is $PSU(N) = SU(N)/\mathbb{Z}_N$. As commented earlier, the quotient by $\mathbb{Z}_N$ follows from the fact that every gauge invariant operator transforms trivially under the center of $PSU(N)$. Hence, we can couple the system to classical background $PSU(N)$ gauge fields.

We denote the $PSU(N)$ bundle by $\mathcal{P}$. It is characterized by the second Stiefel-Whitney class $w_2(\mathcal{P})$, which is an integer modulo $N$. As in the $\mathbb{CP}^{N-1}$ model, for $PSU(N)$ background fields

that are also $SU(N)$ bundles (i.e. $w_2(\mathcal{P}) = 0$) the $N$ gauge fields $a_I$ are ordinary $U(1)$ gauge fields satisfying $\oint \frac{da_I}{2\pi} \in \mathbb{Z}$. But for nonzero $w_2(\mathcal{P})$ the $U(1)$ the gauge fields have

$$\oint \frac{da_I}{2\pi} = \frac{w_2(\mathcal{P})}{N} \quad \text{mod } 1 \,. \tag{2.16}$$

The introduction of these background fields allows us to add to the Lagrangian a counterterm

$$\frac{2\pi}{N} p w_2(\mathcal{P}) \,, \tag{2.17}$$

with $p$ an integer modulo $N$. This integer $p$ does not affect any local physics in the bulk.

The model is characterized by $N$ angles $\theta_I$ as in (2.7). Each is $2\pi$ periodic. However, when $w_2(\mathcal{P})$ is nonzero the fluxes of $a_I$ are fractional multiples of $2\pi$ (see (2.16)) and $\theta_I$ are not quite $2\pi$ periodic [12]. As in the $\mathbb{CP}^{N-1}$ model, a shift of any $\theta_I$ by $2\pi$ does not leave the theory invariant, but shifts $p$ by 1. Therefore, the theory is characterized by $(\theta_1, \theta_2, ..., \theta_N, p)$ with the identifications

$$(\theta_1, \theta_2, ..., \theta_N, p) \sim (\theta_1 + 2\pi, \theta_2, ..., \theta_N, p+1) \sim (\theta_1, \theta_2 + 2\pi, ..., \theta_N, p+1) \sim ...$$
$$\sim (\theta_1, \theta_2, ..., \theta_N, p+N) \,. \tag{2.18}$$

Physically, $p$ labels the $N$-ality of the $PSU(N)$ representation on a boundary. Shifting any $\theta_I$ by $2\pi$ leads to a pair creation of $\phi^I$ particles to screen it, but since they transform nontrivially under $PSU(N)$, this pair creation changes the $N$-ality of the representation on the boundary and leads to (2.18).

# 3 Discrete Global Symmetry and 't Hooft Anomaly

In addition to the continuous $PSU(N)$ global symmetry, the $SU(N)/U(1)^{N-1}$ sigma model (2.12) has various discrete global symmetry at special loci on the parameter space of $r_I, \theta_I, G_{IJ}, B_{IJ}$. In this section we analyze these global symmetries and their 't Hooft anomalies. Our discussion on the 't Hooft anomaly will follow [12].

## 3.1 $\mathbb{S}_N$ Symmetry

Consider the $\mathbb{S}_N$ symmetry that permutes the index $I$. We will be mostly interested in the case when this $\mathbb{S}_N$ symmetry is explicitly broken by generic values of the parameters $r_I, G_{IJ}, \theta_I, B_{IJ}$, but it would still be convenient to organize the couplings using this broken symmetry.

Let $\mathbf{N-1}$ be the standard representation of $\mathbb{S}_N$ associated to the partition $(1, N-1)$.

The parameters $r_I$ and $G_{IJ}$ are subject to redundancies we discussed above. The set of distinct parameters transform in the symmetric product representation of two $(\mathbf{N-1})$ (see, for example, [13]):[7]

$$Sym^2(\mathbf{N-1}) = \mathbf{1} \oplus (\mathbf{N-1}) \oplus \frac{\mathbf{N(N-3)}}{\mathbf{2}}, \tag{3.1}$$

where $\frac{\mathbf{N(N-3)}}{\mathbf{2}}$ is the $\mathbb{S}_N$ irrep associated to the partition $(2, N-2)$. The $r_I$'s can be invariantly identified as the parameters in the $\mathbf{1} \oplus (\mathbf{N-1})$ representation, while the remaining $G_{IJ}$'s can

---

[7]In our convention the trivial $\mathbb{S}_N$ representation corresponds to the partition $(N)$, while the one-dimensional sign representation corresponds to the partition $(1^N)$.

be identified as the $\frac{N(N-3)}{2}$ part. In particular, the trivial representation **1** corresponds to the overall size of the flag manifold.

The distinct parameters $\theta_I$ and $B_{IJ}$ transform in the antisymmetric product representation of two $\mathbf{1} \oplus (\mathbf{N-1})$:

$$\Lambda^2(\mathbf{1} \oplus (\mathbf{N-1})) = (\mathbf{N-1}) \oplus \frac{(\mathbf{N-1})(\mathbf{N-2})}{2}, \tag{3.2}$$

where $\frac{(N-1)(N-2)}{2} = \Lambda^2(\mathbf{N-1})$ corresponds to the partition $(1^2, N-2)$. The $\theta_I$'s can be invariantly identified as the parameters in the $\mathbf{N-1}$ representation, while the remaining $B_{IJ}$'s correspond to the $\frac{(N-1)(N-2)}{2}$ representation.

## 3.2 $\mathbb{Z}_N$ Symmetry

Below we will be interested in the $\mathbb{Z}_N$ global symmetry that cyclically shifts

$$\mathbb{Z}_N : \quad \phi_i^I \to \phi_i^{I+1}, \qquad a_I \to a_{I+1}, \tag{3.3}$$

with $I = N$ maps to $I = 1$. Let us determine the conditions on the parameters so that the theory is invariant under this $\mathbb{Z}_N$ symmetry. Clearly we need

$$\begin{aligned} r_I &= r, \\ \theta_I &= \theta_0 + n\frac{2\pi I}{N}, \quad I = 1, 2, \cdots, N, \end{aligned} \tag{3.4}$$

where $n = 0, 1, 2, \cdots, N-1$. Note that two configurations labeled by $n$ and $n'$ with $gcd(n, N) = gcd(n', N)$ are related by a field redefinition and will not be distinguished.

When we turn on a nontrivial $PSU(N)$ background, we can add to the Lagrangian a counterterm (2.17) with $p$ an integer modulo $N$. Let us study the $\mathbb{Z}_N$ global symmetry when this counterterm is taken into account. We start with the $\mathbb{Z}_N$ symmetric $\theta$ angles labeled by $n$:

$$\begin{aligned} &\left( \theta_1 = \frac{2\pi n}{N}, \theta_2 = \frac{4\pi n}{N}, \cdots, \theta_{N-1} = \frac{2\pi(N-1)n}{N}, \theta_N = 0, p \right) \\ &\underset{\mathbb{Z}_N}{\to} \left( \theta_1 = \frac{4\pi n}{N}, \theta_2 = \frac{6\pi n}{N}, \cdots, \theta_{N-1} = 0, \theta_N = \frac{2\pi n}{N}, p \right) \\ &\to \left( \theta_1 = \frac{2\pi n}{N}, \theta_2 = \frac{4\pi n}{N}, \cdots, \theta_{N-1} = -\frac{2\pi n}{N}, \theta_N = 0, p \right) \\ &\sim \left( \theta_1 = \frac{2\pi n}{N}, \theta_2 = \frac{4\pi n}{N}, \cdots, \theta_{N-1} = 2\pi n - \frac{2\pi n}{N}, \theta_N = 0, p+n \right). \end{aligned} \tag{3.5}$$

Let us explain each step. In the second line we perform the $\mathbb{Z}_N$ symmetry transformation on the gauge fields $a_I$, whose effect is equivalent to changing the $\theta$ angles as given in the second line. In the third line we use (2.15) to shift all the $\theta$ angles simultaneously by $-2\pi n/N$. In the forth line we shift only $\theta_{N-1}$ by $2\pi n$, at the price of shifting $p$ by $n$ at the same time. We see that the previous $\mathbb{Z}_N$ symmetric configuration is no longer invariant when the counterterm (2.17) is taken into account. Instead, the parameters are shifted as

$$\left( \theta_I = \frac{2\pi I n}{N}, p \right) \to \left( \theta_I = \frac{2\pi I n}{N}, p+n \right). \tag{3.6}$$

The shift (3.6) means that at the $\mathbb{Z}_N$ invariant point (3.4) there is a mixed 't Hooft anomaly between the $PSU(N)$ and the $\mathbb{Z}_N$ global symmetries [4], labeled by an integer $n$ modulo $N$.[8]

---

[8]A similar thing happens in the $\mathbb{CP}^{N-1}$ model. There this discrete $\mathbb{Z}_N$ symmetry is replaced by a $\mathbb{Z}_2^{charge}$ charge conjugation symmetry, which is present at $\theta = 0, \pi$. And there is a nontrivial mixed anomaly between the global $PSU(N)$ and this $\mathbb{Z}_2^{charge}$ symmetry at $\theta = \pi$. It leads to the conclusion that the IR physics must be nontrivial at $\theta = \pi$. See Section 7.2 for more details.

This anomaly can be represented as the three-dimensional term

$$\frac{2\pi n}{N} \int A \cup w_2(\mathcal{P}) \,, \tag{3.7}$$

where $\mathcal{P}$ is the background $PSU(N)$ bundle and $A$ is a background $\mathbb{Z}_N$ gauge field. Physically, it means that the action of the global $\mathbb{Z}_N$ symmetry must be accompanied with changing the $N$-ality of the $SU(N)$ representation at the boundary by $n$ units. This anomaly means that at these values of $\theta$ the IR system cannot be trivial. Either there is a first order transition associated with the spontaneous breaking of this $\mathbb{Z}_N$, or there is a nontrivial fixed point there.

### 3.2.1 $\mathbb{Z}_N$ Invariant Deformations

We have studied the constraints on the original Lagrangian (2.7) by the $\mathbb{Z}_N$ symmetry. Let us proceed to study the $\mathbb{Z}_N$ invariant two-derivative deformations (2.11). Let us define

$$\mathcal{G}_{IJ} \equiv \delta^{\mu\nu} \left( \sum_i \phi_i^J \partial_\mu \bar{\phi}_I^i \right) \left( \sum_j \bar{\phi}_J^j \partial_\nu \phi_j^I \right) \,, \tag{3.8}$$

$$\mathcal{B}_{IJ} \equiv \epsilon^{\mu\nu} \left( \sum_i \phi_i^J \partial_\mu \bar{\phi}_I^i \right) \left( \sum_j \bar{\phi}_J^j \partial_\nu \phi_j^I \right) \,. \tag{3.9}$$

We will first count the $\mathbb{Z}_N$ invariant $G$ deformations. When $N$ is even, we have $N/2$ such deformations. Each one of them can be obtained by starting from $\mathcal{G}_{1J}$ with $J = 2, \cdots, N/2 + 1$ and adding its $\mathbb{Z}_N$ images. Explicitly, they are

$$
\begin{aligned}
&\mathcal{G}_{12} + \mathcal{G}_{23} + \cdots + \mathcal{G}_{N1} \,, \\
&\mathcal{G}_{13} + \mathcal{G}_{24} + \cdots + \mathcal{G}_{N2} \,, \\
&\;\;\vdots \\
&\mathcal{G}_{1\frac{N}{2}} + \mathcal{G}_{2\frac{N+2}{2}} + \cdots + \mathcal{G}_{N\frac{N-2}{2}} \,, \\
&\mathcal{G}_{1\frac{N+2}{2}} + \mathcal{G}_{2\frac{N+4}{2}} + \cdots + \mathcal{G}_{\frac{N}{2}N} \,, \quad (N : \text{ even}) \,.
\end{aligned} \tag{3.10}
$$

Note that the last term is special; it only has $N/2$ terms, whereas the other terms have $N$ terms. When $N$ is odd, we can start with $\mathcal{G}_{1J}$ with $J = 2, \cdots, (N+1)/2$ and add its $\mathbb{Z}_N$ images:

$$
\begin{aligned}
&\mathcal{G}_{12} + \mathcal{G}_{23} + \cdots + \mathcal{G}_{N1} \,, \\
&\;\;\vdots \\
&\mathcal{G}_{1\frac{N+1}{2}} + \mathcal{G}_{2\frac{N+3}{2}} + \cdots + \mathcal{G}_{N\frac{N-1}{2}} \,, \quad (N : \text{ odd}) \,.
\end{aligned} \tag{3.11}
$$

Hence, there are $\lfloor \frac{N}{2} \rfloor$ $\mathbb{Z}_N$ invariant $G$ deformations.

Moving on to the $B$ deformations. The analysis is almost identical except that the last term in (3.10) becomes 0 when we add up all the $\mathbb{Z}_N$ images, due to the antisymmetric property of $\mathcal{B}_{IJ}$. It follows that there are only $(N-2)/2$ $B$ deformation terms when $N$ is even, while there are still $(N-1)/2$ terms when $N$ is odd.

To summarize, there are $N(N-1)/2$ $G$ deformations and $N(N-1)/2$ $B$ deformations, before imposing the $\mathbb{Z}_N$ symmetry. Imposing the $\mathbb{Z}_N$ symmetry, there are $\lfloor \frac{N}{2} \rfloor$ $\mathbb{Z}_N$ invariant $G$ deformations and $\lfloor \frac{N-1}{2} \rfloor$ $\mathbb{Z}_N$ invariant $B$ deformations.

### 3.3 $\mathbb{Z}_2$ Symmetries

For special choices of the parameters, there are various enhanced $\mathbb{Z}_2$ global symmetries. We will analyze these $\mathbb{Z}_2$ symmetries and their anomaly.[9]

Consider the following $\mathbb{Z}_2^C$ charge conjugation action:

$$\mathbb{Z}_2^C: \quad \phi_i^I \to \bar{\phi}_{N-I+1}^i, \qquad a_I \to -a_{N-I+1}. \tag{3.12}$$

It is a global symmetry if the parameters $r_I, \theta_I, G_{IJ}, B_{IJ}$ are chosen to be $\mathbb{Z}_N$ symmetric as discussed in Section 3.2. In particular, $\mathbb{Z}_2^C$ is preserved only if the $\theta$-angles are given as in (3.4), i.e. $\theta_I = n\frac{2\pi I}{N}$ for some integer $n$.

Is there a mixed anomaly between $\mathbb{Z}_2^C$ and the $PSU(N)$ symmetry? To settle this, we turn on a $PSU(N)$ background and add to the Lagrangian a counterterm $\frac{2\pi}{N} p w_2(\mathcal{P})$ (2.17). Next, we ask whether the $\mathbb{Z}_N$ symmetric $\theta$-angles are still invariant under $\mathbb{Z}_2^C$ when the counterterm is taken into account:

$$
\begin{aligned}
&\left(\theta_1 = \frac{2\pi n}{N}, \theta_2 = 2\frac{2\pi n}{N}, \cdots, \theta_{N-1} = (N-1)\frac{2\pi n}{N}, \theta_N = 0, p\right) \\
\underset{\mathbb{Z}_2^C}{\to}\ &\left(\theta_1 = 0, \theta_2 = -(N-1)\frac{2\pi n}{N}, \cdots, \theta_{N-1} = -2\frac{2\pi n}{N}, \theta_N = -\frac{2\pi n}{N}, -p\right) \\
\to\ &\left(\theta_1 = \frac{2\pi n}{N}, \theta_2 = -(N-2)\frac{2\pi n}{N}, \cdots, \theta_{N-1} = -\frac{2\pi n}{N}, \theta_N = 0, -p\right) \\
\sim\ &\left(\theta_1 = \frac{2\pi n}{N}, \theta_2 = 2\frac{2\pi n}{N}, \cdots, \theta_{N-1} = 2\pi n - \frac{2\pi n}{N}, \theta_N = 0, -p + n(N-2)\right) \\
\sim\ &\left(\theta_1 = \frac{2\pi n}{N}, \theta_2 = 2\frac{2\pi n}{N}, \cdots, \theta_{N-1} = (N-1)\frac{2\pi n}{N}, \theta_N = 0, -p - 2n\right).
\end{aligned}
\tag{3.13}
$$

We can choose a counterterm $p = -n$ such that the configuration $(\theta_I, p)$ returns to itself under $\mathbb{Z}_2^C$, so there is no mixed anomaly between $PSU(N)$ and $\mathbb{Z}_2^C$ in the flag sigma model $SU(N)/U(1)^{N-1}$.[10] The same conclusion was also arrived in [4] via a different computation.

We can consider another charge conjugation $\mathbb{Z}_2^{charge}$ action

$$\mathbb{Z}_2^{charge}: \quad \phi_i^I \to \bar{\phi}_I^i, \qquad a_I \to -a_I. \tag{3.14}$$

It is a symmetry if all the $\theta_I$'s are either 0 or $\pi$. Without loss of generality consider the point in the parameter space

$$
\theta_I = \begin{cases} \pi & \text{for } I = 1, ..., L \\ 0 & \text{for } I = L+1, ..., N-1. \end{cases}
\tag{3.15}
$$

Then the $\mathbb{Z}_2^{charge}$ charge conjugation symmetry acts as

$$(\theta_I, p) \to (-\theta_I, -p) \sim (\theta_I, -p + L), \tag{3.16}$$

---

[9]Some of these $\mathbb{Z}_2$'s will be referred as charge conjugation. However, the notion of charge conjugation is not invariant under field redefinition nor unique. For example, we can always combine a charge conjugation $\mathbb{Z}_2$ with another $\mathbb{Z}_2$ global symmetry. The combined $\mathbb{Z}_2$ can also be called the charge conjugation. Furthermore, the charge conjugation in one presentation of the model might become a symmetry that does not involve any complex conjugation in another presentation (see Section 7.3 for example). Below we will choose to call some $\mathbb{Z}_2$'s the charge conjugation simply because they involve complex conjugation of the $\phi$ field, but the reader should not assign any invariant meaning to this terminology.

[10]In the $N = 2$ case, this $\mathbb{Z}_2^C$ is an element of $PSU(2)$. On the other hand, there is a mixed anomaly in the $\mathbb{CP}^1$ model between $PSU(2)$ and the $\mathbb{Z}_{N=2}$ discussed in (3.3). Below in Section 7.3 we give a detailed discussion on the global symmetry in the $\mathbb{CP}^1$ sigma model.

where we used (2.18).

When $L$ is odd and $N$ is even, there is no choice of $p$ such that the above configuration is $\mathbb{Z}_2^{charge}$ invariant, so there is a mixed anomaly between the continuous $PSU(N)$ global symmetry and $\mathbb{Z}_2^{charge}$. This anomaly can be represented by the $3d$ term

$$\frac{2\pi L}{2} \int C \cup w_2(\mathcal{P}) \,, \tag{3.17}$$

where $\mathcal{P}$ is the background $PSU(N)$ bundle and $C$ is a background $\mathbb{Z}_2^{charge}$ gauge field. (Note that this is meaningful only for even $N$ and is nontrivial only for odd $L$.) Physically, the symmetry action involves a shift of $\theta_I$ with $I = 1,...,L$ by $2\pi$ and this leads to $L$ pair creations of $\phi^I$ quanta, which move to the boundary. Consequently, the $N$-ality of the representation on the boundary changes by $L$ units. This anomaly means that except for $L = 0$, the long distance physics at these points cannot be trivial. Either the system is gapless there, or this discrete symmetry is spontaneously broken. (The continuous $PSU(N)$ symmetry cannot be broken because we are in two dimensions.)

If on the other hand $L$ and $N$ are both odd, we can choose a $p$ such that the above configuration is $\mathbb{Z}_2^{charge}$ invariant. Similarly, when $L$ is even there is also such a choice of $p$ to preserve the $\mathbb{Z}_2^{charge}$ symmetry. In these cases there is no mixed anomaly between $PSU(N)$ and $\mathbb{Z}_2^{charge}$.

Our system also has a $CP$ symmetry at any $\theta$, so the discussion in this section can be stated equivalently as associated with a parity transformation rather than charge conjugation. $CP$ acts as

$$\phi_i^I(x,t) \to \bar{\phi}_I^i(-x,t), \qquad a_{I\mu}(x,t) \to -(-1)^\mu a_{I\mu}(-x,t), \qquad \mu = 0,1 \,. \tag{3.18}$$

Note that both the $\mathbb{Z}_N$ invariant $G$ and $B$ deformations are even under the $\mathbb{Z}_2^C$ defined in (3.12). On the other hand, the $G$ and $B$ deformations are even and odd under $\mathbb{Z}_2^{charge}$, respectively:

$$\begin{aligned} \mathbb{Z}_2^{charge} : \mathcal{G}_{IJ} &\to +\mathcal{G}_{IJ} \,, \\ \mathcal{B}_{IJ} &\to -\mathcal{B}_{IJ} \,. \end{aligned} \tag{3.19}$$

This can also be seen by noting that both the $G$ and $B$ deformations are $CP$ invariant but the former is $P$ even while the latter is $P$ odd. Hence, the $G$ deformation is $C$ even and the $B$ deformation is $C$ odd.

# 4 Deformation of the WZW Model

The $SU(N)/U(1)^{N-1}$ sigma model with special parameters admits an alternative description in terms of the $SU(N)$ WZW model deformed by certain potential terms (top arrow in Figure 1). In particular, we will restrict to the $\mathbb{Z}_N$ symmetric choice of parameters discussed in Section 3.2. The global symmetry of the flag sigma model is then

$$G_{UV} = (PSU(N) \times \mathbb{Z}_N) \rtimes \mathbb{Z}_2^C \,. \tag{4.1}$$

$G_{UV}$ is embedded into the global symmetry $G_{WZW}$ of the WZW model via the deformation, and they share the same anomaly. This embedding makes it manifest that the flow we want to explore is at least kinematically possible as far as the anomaly is concerned. The embedding of the $\mathbb{CP}^1$ sigma model (which is the $N = 2$ case of the flag sigma model) into the $SU(2)_1$ WZW

model was discussed in [3]. The general $N$ case was first discussed in [4]. Our discussion is slightly different. The anomaly in the $SU(N)$ WZW model was discussed in [4, 14, 15].

Let $U \in SU(N)$ be the fundamental field in the WZW model. The UV WZW model is the group manifold sigma model plus a WZ term:

$$\frac{R}{2} \int_{M_2} \text{Tr}[\partial_\mu U \partial^\mu U^\dagger] + \frac{\text{i}}{12\pi} k \int_{M_3} \text{Tr}[(U^\dagger \text{d}U)^3], \tag{4.2}$$

where $M_2$ is the two dimensional spacetime and $M_3$ is a three-manifold whose boundary is $M_2$, i.e. $\partial M_3 = M_2$. The coefficient $k$ is quantized to be a positive integer. In the UV, the coupling constant $R$, which is the square of the size of the target space, is large and the theory is approximately $N^2 - 1$ free bosons. As we flow to the IR, the coupling $R$ decreases and eventually hit a fixed point at $R = \frac{k}{4\pi}$, which defines the WZW CFT [2].

## 4.1 Global Symmetry

Let us analyze the global symmetry of the WZW model. For generic $N$, the global symmetry is $\frac{SU(N)_L \times SU(N)_R}{\mathbb{Z}_N} \rtimes \mathbb{Z}_2^C$, where the $\mathbb{Z}_2^C$ acts as $U \to U^*$. The $\frac{SU(N)_L \times SU(N)_R}{\mathbb{Z}_N}$ flavor symmetry acts on $U$ as $U \to V_L U V_R^\dagger$. Here $V_L \in SU(N)_L$ and $V_R \in SU(N)_R$ and they are subject to the identification $(V_L, V_R) \sim (V_L \omega, V_R \omega)$ with $\omega = e^{2\pi \text{i}/N}$. We will pay special attention to the subgroup $(PSU(N) \times \mathbb{Z}_N) \rtimes \mathbb{Z}_2^C$, where $PSU(N)$ is the diagonal subgroup with $V_L = V_R$ and the $\mathbb{Z}_N$ factor acts on $U$ as $U \to \omega U$.

The global symmetry for $N = 2$ is different. The global symmetry of the $SU(2)$ WZW model is $\frac{SU(2)_L \times SU(2)_R}{\mathbb{Z}_2}$, which contains a subgroup $PSU(2) \times \mathbb{Z}_2$. The $\mathbb{Z}_2$ acts on the WZW fundamental field $U$ as $U \to -U$. Notice that $U \to U^*$ is included in $PSU(2)$.[11]

To summarize, the global symmetry $G_{WZW}$ of the $SU(N)$ WZW model is

$$SU(2): \quad G_{WZW} = \frac{SU(2)_L \times SU(2)_R}{\mathbb{Z}_2} \supset PSU(2) \times \mathbb{Z}_2, \tag{4.3}$$

$$SU(N): \quad G_{WZW} = \frac{SU(N)_L \times SU(N)_R}{\mathbb{Z}_N} \rtimes \mathbb{Z}_2^C \supset (PSU(N) \times \mathbb{Z}_N) \rtimes \mathbb{Z}_2^C, \quad N > 2, \tag{4.4}$$

where we have highlighted a particular subgroup on the right that will be of importance.

Let us also translate the action of the two $\mathbb{Z}_2$ symmetries (3.12) and (3.18) discussed in Section 3.3 in terms of the WZW fundamental field $U$:

$$\mathbb{Z}_2^C: \ U(x,t) \to U(x,t)^*, \quad CP: \ U(x,t) \to U(-x,t)^T. \tag{4.5}$$

## 4.2 Deformation to the Flag Sigma Model

We now discuss the deformation of the UV WZW model to the flag sigma model, illustrated in the top line in Figure 1. The main point is that the flag manifold is a subspace of the WZW model target space $SU(N)$. We look for a potential as a function of $U$, which is invariant under $G_{UV}$ and restricts $U$ to take values in that subspace.[12] Starting from the UV WZW model (4.2),

---

[11] To see this, let us parameterize the fundamental field $U \in SU(2)$ as $U = \begin{pmatrix} a & b \\ -\bar{b} & \bar{a} \end{pmatrix}$, $|a|^2 + |b|^2 = 1$. The action $U \to U^*$ is included in $PSU(2)$ as $U \to \begin{pmatrix} 0 & 1 \\ -1 & 0 \end{pmatrix} U \begin{pmatrix} 0 & -1 \\ 1 & 0 \end{pmatrix} = U^*$.

[12] Generally, we can restrict the field of the sigma model on a manifold $Y$ with isometry $G_Y$ to take value in a submanifold $X \subset Y$ by turning on a potential $V$ on $Y$ satisfying $V|_X = 0$ and $V|_{Y \setminus X} > 0$. For example, we can choose $V$ to be a positive constant away from $X$, and smoothly interpolate to zero on $X$. Furthermore, we need the potential to be invariant under the subgroup $G_X$ of $G_Y$ that stabilizes $X$. This can be achieved by averaging $V$ over the action of $G_X$. This discussion guarantees that we can always find an appropriate invariant potential. The construction (4.6) is a concrete realization of such a potential.

we turn on the potential:

$$\sum_{n=1}^{\lfloor N/2\rfloor} g_n \operatorname{Tr}[U^n]\operatorname{Tr}[(U^\dagger)^n]. \tag{4.6}$$

Note that we only sum the power of $U$ up to $\lfloor N/2\rfloor$, instead of $N-1$.[13] This potential term preserves the diagonal $PSU(N)$ as well as the $\mathbb{Z}_N$ symmetry. If we send $g_n \to +\infty$, the above potential restricts the fundamental field $U$ to satisfy

$$\operatorname{Tr}[U^n] = 0, \quad n = 1,2,3,\cdots,N-1. \tag{4.7}$$

It is obvious that the traces with $n = 1,2,\cdots,\lfloor N/2\rfloor$ are forced to vanish by the potential. In Appendix A we further show that this also implies the vanishing of $\operatorname{Tr}[U^n]$ with $n = \lfloor N/2\rfloor + 1,\cdots,N-1$. For any such $U$, the characteristic polynomial reduces to

$$\det(\lambda I - U) = \lambda^N - \frac{1}{N}\operatorname{Tr}[U^N]. \tag{4.8}$$

It follows that the $N$ eigenvalues are the distinct $N$-th roots of unity multiplied by an overall constant. The overall constant is fixed by $\det U = 1$ so that the eigenvalues are:

$$\omega^{-(N-1)/2}(1,\omega,\omega^2,\cdots,\omega^{N-1}). \tag{4.9}$$

To conclude, we turn on the potential (4.6) to restrict the WZW fundamental field $U$ to satisfy (4.7). This means that

$$\begin{aligned}
U &= \phi\,\Omega_0\,\phi^\dagger, \\
\Omega_0 &= \omega^{-(N-1)/2}\operatorname{diag}(1,\omega,\omega^2,\cdots,\omega^{N-1}),
\end{aligned} \tag{4.10}$$

where $\omega = e^{2\pi i/N}$ and $\phi \in U(N)$. More explicitly, $U_i{}^j = \sum_{I,J}\phi_i^I(\Omega_0)_I{}^J\bar\phi_J^j$. There are redundancies in this parametrization: two different $\phi$'s might give identical $U$, and should therefore be identified. The redundancy is $U(1)^N$ which acts on $\phi$ as

$$U(1)^N: \quad \phi \to \phi\,\operatorname{diag}(e^{i\alpha_1},\cdots,e^{i\alpha_N}). \tag{4.11}$$

Hence, the distinct $\phi$'s take values in $U(N)/U(1)^N$. We identify these $\phi$'s with the fields in the description of the model presented in Section 2.1.

Let us discuss how the global symmetries act both in terms of the flag sigma model field $\phi$ and the WZW fundamental field $U$. The $PSU(N)$ symmetry acts on $\phi_i^I$ as $\phi_i^I \to V_i^j\phi_j^I$ with $V \in PSU(N)$. The WZW fundamental field $U$ is related to $\phi$ as $U = \phi\Omega_0\phi^\dagger$. Hence, the $PSU(N)$ symmetry indeed translates into the diagonal subgroup of $\frac{SU(N)_L \times SU(N)_R}{\mathbb{Z}_N}$ in the WZW model that acts on $U$ as $U \to VUV^\dagger$.

The $\mathbb{Z}_N$ global symmetry, on the other hand, acts on the $\phi$'s as cyclic permutation (3.3), while it acts on $U$ as $\mathbb{Z}_N: U \to \omega U$.

In Appendix B we will enumerate the number of $PSU(N)$ and $PSU(N) \times \mathbb{Z}_N$ invariant deformations of the flag sigma model in terms of the WZW fundamental field $U$. This reproduces the counting from the original Lagrangian (2.12) in terms of $\phi_i^I$ in Section 2.1 and in Section 3.2.1.

---

[13]Unlike [4], for $N > 3$ we extend the sum beyond $n = 1$.

### 4.2.1 The WZW Action

Below we will substitute (4.10) into the WZW Lagrangian (4.2) and rewrite the Lagrangian in terms of $\phi_i^I$. In particular we will discuss how the Wess-Zumino term in the WZW model reduces to the $\theta$-angle terms plus the $B$ deformation terms. This was done in [4] and we repeat the calculation here for readers' convenience.

In this section we will assume a more general $\Omega_0$ matrix than the one (4.10) that is relevant for the $\mathbb{Z}_N$ symmetric flag manifold $SU(N)/U(1)^{N-1}$. We will take

$$\Omega_0 = \text{diag}(e^{i\varphi_1}, e^{i\varphi_2}, \cdots, e^{i\varphi_N}). \tag{4.12}$$

For the $SU(N)/U(1)^{N-1}$ case, $\varphi_I = -2\pi\frac{N+1}{2N} + 2\pi\frac{I}{N}$. This generalization will be useful in Section 7 when we talk about more general flag manifolds.

Using (4.10), the kinetic term of the WZW action (4.2) can be easily computed as:

$$\frac{R}{2}\text{Tr}[\partial_\mu U \partial^\mu U^\dagger] = R\sum_{I,i}\partial_\mu\phi_i^I\partial^\mu\bar\phi_I^i - R\sum_{I,J}e^{i\varphi_I - i\varphi_J}\left(\sum_i\phi_i^J\partial_\mu\bar\phi_I^i\right)\left(\sum_j\bar\phi_J^j\partial^\mu\phi_j^I\right). \tag{4.13}$$

Let us proceed to compute the WZ term. We will take $M_3 = M_2 \times \mathcal{I}$ where $\mathcal{I} = [0,1]$ is an interval. This choice of the three-manifold will prove to be computationally convenient. However, the boundary of $M_3$ consists of two copies of $M_2$, instead of one. To remedy this, we will consider a field extension such that only one of the two boundaries contributes. Let us denote the coordinates on $M_2$ by $z, \bar z$, and the coordinate on $\mathcal{I}$ as $y \in [0,1]$. Since the WZ action (after exponentiation) doesn't depend on the extension, we will proceed our calculation with a particular choice:

$$U(z, \bar z, y) = \phi(z, \bar z)\Omega(y)\phi(z, \bar z)^\dagger, \tag{4.14}$$

where

$$\begin{aligned}\Omega(y) &= \text{diag}(e^{i\varphi_1(y)}, e^{i\varphi_2(y)}, \cdots, e^{i\varphi_N(y)}), \\ \varphi_I(0) &= \varphi_I, \quad \varphi_I(1) = 0,\end{aligned} \tag{4.15}$$

such that $\Omega(0) = \Omega_0$, $\Omega(1) = I$, and $\Omega(y)^\dagger\Omega(y) = I$. We will also extend $\Omega(y)$ in a way that $\det\Omega(y) = 1$ so that $U(z, \bar z, y)$ is in $SU(N)$. Note that even though such an extension of $U$ does not minimize the potential (4.6) in the bulk of $M_3$, it does not affect the WZ action since the latter is insensitive to the extension as long as the boundary values are unchanged. The extended $U$ has the property that at $y = 0$, it reduces to the original field configuration (4.10), while on the other end it reduces to the identity matrix:

$$\begin{aligned}U(z, \bar z, y = 0) &= U(z, \bar z) = \phi\Omega_0\phi^\dagger, \\ U(z, \bar z, y = 1) &= I.\end{aligned} \tag{4.16}$$

Since $U$ reduces to the identity matrix on the other end $y = 1$, the WZ term will only receive contribution from one copy of $M_2$ located at $y = 0$. Another way to say this is that since $U(z, \bar z, y = 1)$ is a constant for all $z, \bar z$, we can effectively compactify that boundary to a point.

Let us proceed to the actual calculation by substituting (4.14) into the second term of (4.2). First we note that $U^\dagger dU = \phi\Omega^\dagger\phi^\dagger d\phi\Omega\phi^\dagger + \phi\Omega^\dagger d\Omega\phi^\dagger + \phi d\phi^\dagger$. To compute the WZ term, since only the factor $\Omega(y)$ depends on the $y$-direction, we need to have exactly one factor of $d\Omega$ when taking the cubic power of $U^\dagger dU$. We have

$$\begin{aligned}\text{Tr}[(U^\dagger dU)^3] = 3\text{Tr}[&-\Omega^\dagger d\Omega d\phi^\dagger d\phi - d\Omega\Omega^\dagger d\phi^\dagger d\phi \\ &-d\Omega\phi^\dagger d\phi\Omega^\dagger\phi^\dagger d\phi + d\Omega^\dagger\phi^\dagger d\phi\Omega\phi^\dagger d\phi].\end{aligned} \tag{4.17}$$

The first two terms give us the $\theta$-terms at the $\mathbb{Z}_N$ symmetric configuration:

$$\frac{\mathrm{i}}{12\pi}(-6ki)\int_{M_2}\sum_{I,i}\epsilon^{\mu\nu}\partial_\mu\bar{\phi}_I^i\partial_\nu\phi_i^I\int_{\varphi_I(1)}^{\varphi_I(0)}\mathrm{d}\varphi_I(y)=\frac{k}{2\pi}\int_{M_2}\sum_I\varphi_I\epsilon^{\mu\nu}\sum_i\partial_\mu\bar{\phi}_I^i\partial_\nu\phi_i^I. \quad (4.18)$$

Note that the WZ term is indeed independent of the extension $\varphi_I(y)$ to the three-manifold $M_3$ but only depends on the boundary values $\varphi_I(0)$. In the case of $SU(N)/U(1)^{N-1}$, $\varphi_I=-2\pi\frac{N+1}{2N}+2\pi\frac{I}{N}$, we obtain the $\mathbb{Z}_N$ symmetric $\theta$ angles in (3.4) with $n=k$:

$$\theta_I=\theta_0+k\frac{2\pi I}{N},\quad I=1,2,\cdots,N, \quad (4.19)$$

where $\theta_0=-2\pi k\frac{N+1}{2N}$. As noted below (3.4), any $n$ with $gcd(n,N)=k$ is related to (4.19) by a field redefinition. Therefore the $\mathbb{Z}_N$-symmetric flag sigma model with any such $n$ can be embedded into the $SU(N)_k$ WZW model.

The last two terms in (4.18) give us the $B$ deformation terms:

$$\frac{\mathrm{i}}{12\pi}(-3ki)\epsilon^{\mu\nu}\sum_{I\neq J}\int_{\varphi_I(1)}^{\varphi_I(0)}\mathrm{d}\varphi_I(y)(e^{\mathrm{i}\varphi_I(y)-\mathrm{i}\varphi_J(y)}+e^{-\mathrm{i}\varphi_I(y)+\mathrm{i}\varphi_J(y)})\sum_{i,j}\int_{M_2}\bar{\phi}_I^i\partial_\mu\phi_i^J\bar{\phi}_J^j\partial_\nu\phi_j^I$$

$$=-\frac{k}{4\pi}\sum_{I\neq J}\sin(\varphi_I(0)-\varphi_J(0))\,\epsilon^{\mu\nu}\int_{M_2}\left(\sum_i\phi_i^J\partial_\mu\bar{\phi}_I^i\right)\left(\sum_j\bar{\phi}_J^j\partial_\nu\phi_j^I\right), \quad (4.20)$$

where we have used $\varphi_I(1)=0$ for all $I=1,\cdots,N$. In the case of $SU(N)/U(1)^{N-1}$, this gives $B_{IJ}=-\frac{k}{4\pi}\sin(\frac{2\pi(I-J)}{N})$. Again the WZ term is independent of how $\varphi_I(y)$'s are extended to $M_3$, but only depends on the boundary values $\varphi_I(0)$.

## 4.3 Symmetry-Preserving Relevant Deformations

In Section 4.2 we have discussed the deformation of the UV WZW model to the flag sigma model and how the global symmetry $G_{UV}$ is embedded into the WZW symmetry $G_{WZW}$ (the top line in Figure 1). This embedding makes the flow from the flag sigma model to the WZW model kinematically possible. Next, we discuss the behavior around the IR WZW point (bottom of Figure 1) and examine the relevant deformations there. If there is no $G_{UV}$-preserving relevant deformations at the WZW CFT, then it is likely that the flow from the sigma model will hit the WZW fixed point without fine-tuning. We will see that it is indeed the case for the proposed flow from the $SU(N)/U(1)^{N-1}$ sigma model with $\theta_I=n\frac{2\pi I}{N}$ and $gcd(n,N)=1$ to the $SU(N)_1$ WZW model.

The WZW CFT has the marginal operator $\sum_{a=1}^{\dim G}j^a\bar{j}^a$, where $j^a$ and $\bar{j}^a$ are the holomorphic and antiholomorphic currents. Other than that, only a current algebra primary operator can be relevant or marginal, since the descendants are necessarily irrelevant. Since we only care about operators that are invariant under $PSU(N)$, the group indices of the current algebra primaries are always understood to be contracted between the left and the right to preserve this diagonal $PSU(N)$.

A current algebra primary of the theory is labeled by a representation $\mathbf{R}$ of $\mathfrak{su}(N)$ and will be denoted by $\mathcal{O}_{\mathbf{R}}$. The allowed representations $\mathbf{R}$ in the $SU(N)_k$ WZW model are those whose sum of the Dynkin labels are less than or equal to $k$.

For $k\geq 2$, the WZW CFT always has a relevant primary operator $\mathcal{O}_{\mathbf{Adj}}$. It is invariant under the center $\mathbb{Z}_N$ and the charge conjugation $\mathbb{Z}_2^C$. Hence, it is a $G_{UV}$-preserving relevant deformation in the $SU(N)_k$ WZW model with $k\geq 2$ [3,16]. The physical consequence is that at least one fine-tuning is necessary to hit the $SU(N)_k$ WZW fixed point with $k\geq 2$ along a

Table 1: Relevant and marginal deformations in the $SU(N)_1$ WZW CFT. Marginal operators are in the square brackets. The operator $\mathcal{O}_{\Lambda^\ell \mathbf{N}}$ is abbreviated as $\mathcal{O}_\ell$, and we have $\overline{\mathcal{O}}_\ell = \mathcal{O}_{N-\ell}$. The third column shows the subgroups of the center $\mathbb{Z}_N$ in (4.3) or (4.4) under which all the relevant and marginal primary operators transform nontrivially.

| $N$ | relevant or marginal $\mathcal{O}_\ell$ | $\mathbb{Z}_r \subseteq \mathbb{Z}_N$ |
|---|---|---|
| $N \leq 7$ | $\mathcal{O}_\ell$ for $1 \leq \ell \leq N-1$ | $\mathbb{Z}_N$ |
| 8 | $\mathcal{O}_1, \mathcal{O}_2, \mathcal{O}_3, [\mathcal{O}_4], \mathcal{O}_5, \mathcal{O}_6, \mathcal{O}_7$ | $\mathbb{Z}_8$ |
| 9 | $\mathcal{O}_1, \mathcal{O}_2, [\mathcal{O}_3], [\mathcal{O}_6], \mathcal{O}_7, \mathcal{O}_8$ | $\mathbb{Z}_9$ |
| $N \geq 10$ | $\mathcal{O}_1, \mathcal{O}_2, \mathcal{O}_{N-2}, \mathcal{O}_{N-1}$ | Any subgroup other than $\mathbb{Z}_2$ and $\{1\}$ |

flow from the flag sigma model. In fact, for $k \geq 2$ and $gcd(k, N) = 1$, it was conjectured that the $SU(N)_k$ WZW model perturbed by $\mathcal{O}_{\mathbf{Adj}}$ flows to the $SU(N)_1$ WZW model in the IR [17].

Let us restrict to the $k = 1$ case because of the above reason. The nontrivial primaries are $\mathcal{O}_{\Lambda^\ell \mathbf{N}}$ for $\ell = 1, 2 \cdots, N-1$, where $\Lambda^\ell \mathbf{N}$ is the $\ell$-th antisymmetric power of the fundamental representation $\mathbf{N}$. In particular, the theory does not have $\mathcal{O}_{\mathbf{Adj}}$. Note that the complex conjugation $\overline{\Lambda^\ell \mathbf{N}}$ of $\Lambda^\ell \mathbf{N}$ is equivalent to $\Lambda^{N-\ell} \mathbf{N}$ and hence $\overline{\mathcal{O}}_{\Lambda^\ell \mathbf{N}} = \mathcal{O}_{\Lambda^{N-\ell} \mathbf{N}}$. The holomorphic conformal weight $h_\ell$ of the primary $\mathcal{O}_{\Lambda^\ell \mathbf{N}}$ is

$$h_\ell = \frac{\ell(N-\ell)}{2N}. \tag{4.21}$$

In the diagonal $SU(N)_1$ WZW CFT, the primary $\mathcal{O}_{\Lambda^\ell \mathbf{N}}$ is relevant if $h_\ell < 1$. The relevant and marginal primary operators in $SU(N)_1$ WZW CFT are listed in Table 1.

Next, we discuss how the $\mathbb{Z}_N$ symmetry acts on these $PSU(N)$-invariant operators. The importance of this $\mathbb{Z}_N$ symmetry in the $SU(N)$ WZW model has been emphasized in [6]. Since the $\mathbb{Z}_N$ generator acts on the fundamental representation $\mathbf{N}$ a phase $e^{2\pi i/N}$, it acts on the rank $\ell$ antisymmetric tensor representation $\Lambda^\ell \mathbf{N}$ by a phase $e^{2\pi i \ell/N}$, so does the corresponding primary $\mathcal{O}_\ell$. Hence, there is no $G_{UV}$-invariant relevant deformation in the $SU(N)_1$ WZW CFT. This suggests that the flag sigma model with $\theta_I = n\frac{2\pi I}{N}$ and $gcd(n, N) = 1$ can flow to the $SU(N)_1$ WZW model without fine-tuning.

In Section 7.1, we will discuss more general flag sigma model whose global symmetry does not contain $\mathbb{Z}_N$, but a subgroup thereof. In Table 1, the subgroups of $\mathbb{Z}_N$ that can be used to exclude all the relevant and marginal operator in the $SU(N)_1$ WZW CFT are also listed.[14] When we further impose the charge conjugation symmetry $\mathbb{Z}_2^C : U \to U^*$, only the sum of a primary and its conjugate is allowed to be turned on.

# 5 Renormalization Group Flow

## 5.1 The General Flow

The $SU(N)$ invariant metric on $SU(N)/U(1)^{N-1}$ can be parameterized by the $r_I$'s and the $G_{IJ}$'s, subject to the redundancy $r_I \to r_I + c_I$ and $G_{IJ} \to G_{IJ} - c_I - c_J$. We can remove the redundancy by setting all the $r_I$'s to be zero. Similarly, the $SU(N)$ invariant $B$-field deformations are

---

[14]When $(N, \ell) = (8, 4), (9, 3), (9, 6)$, the primaries $\mathcal{O}_{\Lambda^\ell \mathbf{N}}$ are marginal (but not exactly marginal). If the marginal deformations are marginally irrelevant in a (codimension zero) region around the CFT point, it is sufficient to only exclude the relevant operators to reach the WZW CFT, and the minimal subgroups are $\mathbb{Z}_4$ and $\mathbb{Z}_3$ for $N = 8$ and 9, respectively.

parameterized by the $\theta_I$'s and the $B_{IJ}$'s, subject to the redundancy $\theta_I \to \theta_I + 2\pi b_I$ combined with $B_{IJ} \to B_{IJ} - b_I - b_J$. We again remove the redundancy by setting all the $\theta_I$'s to be zero.

The $SU(N)$ invariant metric and the $B$-field for the non-linear sigma model on $SU(N)/U(1)^{N-1}$ are then parameterized by $\frac{N(N-1)}{2}$ $G_{IJ}$'s and $\frac{N(N-1)}{2}$ $B_{IJ}$'s, respectively, with $1 \leq I < J \leq N$. In this section we will write down the one-loop beta functions for both $G_{IJ}$ and $B_{IJ}$.

Consider a general non-linear sigma model with Lagrangian

$$\frac{1}{2} \sum_{a,b} (g_{ab}(X)\delta^{\mu\nu} + i b_{ab}(X)\epsilon^{\mu\nu})\partial_\mu X^a \partial_\nu X^b , \tag{5.1}$$

where $g_{ab}(X)$ and $b_{ab}(X)$ are the metric and the $B$-field on the target space, respectively. Here, $a, b, c$ are the indices of the tangent bundle on the target space. $X^a$'s are the (real) coordinates on the target space. The one-loop beta functions of a general non linear sigma model with metric $g_{ab}$ and $b_{ab}$ are [18–20]

$$\frac{\mathrm{d}}{\mathrm{d}\log\mu} g_{ab} = \frac{1}{2\pi} R_{ab} - \frac{1}{8\pi} H_a{}^{cd} H_{bcd} + \cdots ,$$
$$\frac{\mathrm{d}}{\mathrm{d}\log\mu} b_{ab} = -\frac{1}{4\pi} \nabla^c H_{cab} + \cdots , \tag{5.2}$$

where $R_{ab}$ is the Ricci tensor associated to $g_{ab}$, $\nabla$ is the covariant derivative (for the affine connection), and the field strength $H_{abc}$ is defined by

$$H_{abc} = \partial_a b_{bc} + \partial_c b_{ab} + \partial_b b_{ca} . \tag{5.3}$$

The $\cdots$ represent correction from higher-loop contributions in the sigma model. To use these formula, we need to explicitly know the metric $g_{ab}$ and the $b_{ab}$ on the flag manifold as functions of $G_{IJ}$ and $B_{IJ}$. We expand the $\phi$ field as

$$\phi_i^I = 1 + i\Theta_i^I - \frac{1}{2}\sum_k \Theta_k^I \Theta_i^k + i\frac{1}{6}\sum_{k,l} \Theta_k^I \Theta_l^k \Theta_i^l + \mathcal{O}(\Theta^4) . \tag{5.4}$$

$\phi$ being unitary implies that $\Theta$ is hermitian, $(\Theta_I^i)^* = \Theta_i^I$. We use the $U(1)^{N-1}$ gauge

$$\Theta_I^I = 0 \quad \text{for all } I . \tag{5.5}$$

It follows that $\Theta_i^I$ with $I > i$ forms $N(N-1)/2$ complex coordinates of the flag manifold around the origin. A pair $(I, i)$ can be thought as an index of the coordinates $\Theta_i^I$, and its complex conjugate index $\overline{(I, i)}$ is identified with $(i, I)$. Substituting the expansion (5.4) into (2.11) (with $r_I = \theta_I = 0$), we get the linearized Lagrangian

$$\frac{1}{2} \sum_{I \neq i, J \neq j} (g_{(I,i),(J,j)}(\Theta)\delta^{\mu\nu} + b_{(I,i),(J,j)}(\Theta)\epsilon^{\mu\nu})\partial_\mu \Theta_i^I \partial_\nu \Theta_j^J , \tag{5.6}$$

where the the metric $g_{(I,i),(J,j)}(\Theta)$ and the $B$-field $b_{(I,i),(J,j)}(\Theta)$ are functions of the coordinates $\Theta_i^I$ and $G_{IJ}, B_{IJ}$ that can be explicitly computed up to a given order of $\Theta$. Notice that $g_{(I,i),(J,j)} = g_{(i,I),(j,J)}$ and $b_{(I,i),(J,j)} = -b_{(i,I),(j,J)}$.[15] In particular, the metric and the $B$-field at the origin $\Theta_i^I = 0$ is

$$g_{(I,i),(J,j)}(0) = \begin{cases} G_{IJ} & \text{if } i = J, j = I , \\ 0 & \text{otherwise} , \end{cases}$$

$$b_{(I,i),(J,j)}(0) = \begin{cases} -B_{IJ} & \text{if } i = J, j = I , \\ 0 & \text{otherwise} . \end{cases} \tag{5.7}$$

---

[15]Note that $b_{(I,i),(J,j)}$ is real and the second term in (5.6) is purely imaginary in Euclidean signature as it should be. There is no i in front of $b_{(I,i),(J,j)}(\Theta)$ because our coordinates $\Theta_i^I$'s are not real.

To compute the beta functions (5.2) at $\Theta = 0$, it is sufficient to expand $\phi$ up to the third order in $\Theta$, since the beta functions includes two derivatives.[16] For $N = 3, 4$, an explicit computation (with the help of Mathematica) shows

$$R_{(I,J),(J,I)}(\Theta = 0) = N + \frac{1}{2} \sum_{K \neq I,J} \left( \frac{G_{IJ}^2}{G_{IK} G_{JK}} - \frac{G_{IK}}{G_{KJ}} - \frac{G_{JK}}{G_{IK}} \right),$$

$$\sum_{K,k} \nabla^{(K,k)} H_{(K,k),(I,J),(J,I)}(\Theta = 0) = 2 \sum_{K \neq I,J} \frac{G_{IJ}(B_{IJ} - B_{IK} - B_{KJ})}{G_{IK} G_{JK}},$$

$$(5.8)$$

and we conjecture the above form for all $N$. In fact, the Ricci tensor for the $SU(N)$ invariant metric on the flag manifold $SU(N)/U(1)^{N-1}$ was derived in [21] and [22], which agrees with (5.9).

Therefore, from (5.2), (5.7), (5.8), we obtain the one-loop beta functions of $G_{IJ}$ and $B_{IJ}$

$$\frac{\mathrm{d}}{\mathrm{d} \log \mu} G_{IJ} = \frac{1}{2\pi} \left[ N + \frac{1}{2} \sum_{K \neq I,J} \left( \frac{G_{IJ}^2}{G_{IK} G_{JK}} - \frac{G_{IK}}{G_{KJ}} - \frac{G_{JK}}{G_{IK}} \right) \right] + \cdots, \qquad (5.9)$$

$$\frac{\mathrm{d}}{\mathrm{d} \log \mu} B_{IJ} = \frac{1}{2\pi} \sum_{K \neq I,J} \frac{G_{IJ}(B_{IJ} - B_{IK} - B_{KJ})}{G_{IK} G_{JK}} + \cdots. \qquad (5.10)$$

The $\cdots$ in the first line contains the $H^2$ term in the one-loop beta function (5.2). In the following we will study the RG flow in the large volume limit, where the $H^2$ term is suppressed by powers of $1/G$.

## 5.2 $\mathbb{Z}_N$ Invariant Flow of $G_{IJ}$

Let us consider the special case when all the $G_{IJ}$'s are identical:

$$G_{IJ} = G_0, \quad B_{IJ} = 0, \quad \forall \, I, J. \qquad (5.11)$$

This locus preserves the $\mathbb{S}_N$ symmetry that acts on the $I, J$ indices. $G_0$ is the modulus for the overall volume of the manifold.

The Ricci flow equation for $G_0$ on this $\mathbb{S}_N$ invariant locus is

$$\frac{\mathrm{d}}{\mathrm{d} \log \mu} G_0 = \frac{1}{2\pi} \frac{N+2}{2}. \qquad (5.12)$$

The one-loop beta function for $G_0$ is positive as it should be. Indeed, as we go to higher energy, the size $G_0$ grows and the sigma model becomes weakly coupled.

Next, we consider a small perturbation $\delta G_{IJ}$ of $G_{IJ}$ around the $\mathbb{S}_N$ invariant configuration (5.11). To leading order in $\delta G_{IJ}/G_0$, the one-loop beta function for $G_{IJ}$ has an $\mathbb{S}_N$ symmetry which will be used to organize our parameters. As commented in (3.1), the $G$ deformations transform in the $Sym^2(\mathbf{N-1})$ representation of $\mathbb{S}_N$, which can be further decomposed into

$$\mathbf{1} \oplus (\mathbf{N-1}) \oplus \frac{\mathbf{N(N-3)}}{\mathbf{2}}. \qquad (5.13)$$

The trivial irrep $\mathbf{1}$ is the overall volume $G_0$, while the perturbation $\delta G_{IJ}$ can be decomposed into the other two irreducible representations. By the $\mathbb{S}_N$ symmetry, the one-loop beta function for $\delta G_{IJ}$ must take the form

$$\frac{\mathrm{d}}{\mathrm{d} \log \mu} \delta G_{IJ} = g_{\mathbf{R}} \frac{\delta G_{IJ}}{G_0} + \cdots, \quad \delta G_{IJ} \in \mathbf{R} \qquad (5.14)$$

---

[16] In fact it suffices to expand to second order, since the third order can be absorbed by a coordinate change. However we keep the third order here for clarity.

to this order in the expansion of $\delta G_{IJ}/G_0$, where $\mathbf{R}$ is one of the latter two irreps in (5.13). Importantly, the constant $g_{\mathbf{R}}$ only depends on which irrep $\delta G_{IJ}$ belongs to, but not on the specific component of $\delta G_{IJ}$. This symmetry argument greatly simplifies the analysis to determining only two constants $g_{\mathbf{R}}$ at this order. Similarly from the $\mathbb{S}_N$ symmetry, the correction to the flow (5.12) can only arise at order $(\delta G_{IJ}/G_0)^2$.

We would like to further restrict ourselves to the $\mathbb{Z}_N$ (which is a true symmetry non-perturbatively) invariant flows. This amounts to identifying the trivial $\mathbb{Z}_N$ representations in the decomposition of the $\mathbb{S}_N$ irreps in (5.13) into $\mathbb{Z}_N$ irreps.

The trivial $\mathbb{S}_N$ irrep $\mathbf{1}$ obviously descends to the trivial $\mathbb{Z}_N$ irrep, which is the overall volume modulus $G_0$. The standard $\mathbb{S}_N$ irrep $\mathbf{N-1}$, on the other hand, does not contain any $\mathbb{Z}_N$ invariant. It follows that all the remaining $\lfloor \frac{N}{2} \rfloor - 1$ nontrivial $\mathbb{Z}_N$ invariants come from the $\mathbb{S}_N$ irrep $\frac{\mathbf{N(N-3)}}{\mathbf{2}}$. The flow of these $\mathbb{Z}_N$ invariant $\delta G_{IJ}$ is completely determined by a single constant $g_{\frac{\mathbf{N(N-3)}}{2}}$ in (5.14).

We will determine this constant $g_{\frac{\mathbf{N(N-3)}}{2}}$ below. Since all the nontrivial $\mathbb{Z}_N$ invariant $G$ deformations belong to a single irreducible $\mathbb{S}_N$ representation, it suffices to turn on one such deformation and set the others to be zero. For example, we will choose (see Section 3.2.1)

$$
\begin{aligned}
G_{12} = G_{23} = \cdots = G_{N1} = G_0 + \delta G\,, \\
G_{IJ} = G_0 - \frac{2}{N-3}\delta G\,, \quad \text{for all other } G_{IJ}\,,
\end{aligned}
\tag{5.15}
$$

in such a way that the average of $G_{IJ}$ is $G_0$. Let us substitute the above $G_{IJ}$ into the one-loop beta function (5.9) with $I = 1, J = 2$ and expand to leading order in $\delta G/G_0$:

$$
\frac{\mathrm{d}}{\mathrm{d}\log\mu}(G_0 + \delta G) = \frac{1}{2\pi}\frac{N+2}{2} + \frac{1}{2\pi}(N-1)\frac{\delta G}{G_0} + \mathcal{O}\left(\frac{\delta G^2}{G_0^2}\right).
\tag{5.16}
$$

Since the flow (5.12) of $G_0$ is corrected only at the order of $\delta G^2/G_0^2$, we have

$$
\frac{\mathrm{d}}{\mathrm{d}\log\mu}\delta G = \frac{1}{2\pi}(N-1)\frac{\delta G}{G_0} + \cdots\,.
\tag{5.17}
$$

That is, $g_{\frac{\mathbf{N(N-3)}}{2}} = \frac{N-1}{2\pi}$.

The physical coupling is not $\delta G$, but the ratio

$$
\widetilde{\delta G_a} \equiv \frac{\delta G_a}{G_0}\,.
\tag{5.18}
$$

There are several ways to see this combination is the physical coupling constant we should be interested in. From the Lagrangian point of view, this is the coupling constant when we canonically normalize the field. From the geometric point of view, this is the relative "ripple" of the metric normalized with respect to the overall size $G_0$ of the manifold.

The one-loop beta function for the relative $G$ deformation $\widetilde{\delta G} = \delta G/G_0$ is then

$$
\frac{\mathrm{d}}{\mathrm{d}\log\mu}\widetilde{\delta G} = \frac{1}{2\pi}\left(N-1-\frac{N+2}{2}\right)\frac{\widetilde{\delta G}}{G_0} + \cdots = \frac{1}{2\pi}\frac{N-4}{2}\frac{\widetilde{\delta G}}{G_0} + \cdots, \quad (N \geq 4).
\tag{5.19}
$$

Recall that for $N = 3$ the only $\mathbb{Z}_3$ invariant $G$ deformation is the overall volume, and there is no $\delta G$ to talk about. For $N = 4$ the one-loop beta function vanishes at leading order in $\delta G/G_0$. The next contribution will come from two-loop diagrams, which are of order $1/G_0^2$. For $N > 4$, the beta function of $\widetilde{\delta G}$ is negative, meaning to leading order around the large volume point, the "ripple" $\widetilde{\delta G}$ *decreases* when we flow to the IR and hence is irrelevant.

### 5.3   $\mathbb{Z}_N$ **Invariant Flow of** $B_{IJ}$

Consider a small perturbation of $B_{IJ}$ around the $\mathbb{S}_N$ invariant configuration (5.11). We will again use the $\mathbb{S}_N$ symmetry to constrain the flow. As stated in (3.2), the $B$ deformations transform in the $\mathbb{S}_N$ representation:

$$(\mathbf{N-1}) \oplus \frac{(\mathbf{N-1})(\mathbf{N-2})}{\mathbf{2}}. \tag{5.20}$$

The $\mathbb{S}_N$ symmetry constrains the one-loop beta function for $B_{IJ}$ to take the form

$$\frac{\mathrm{d}}{\mathrm{d}\log\mu}B_{IJ} = b_{\mathbf{R}}\frac{B_{IJ}}{G_0} + \cdots, \quad B_{IJ} \in \mathbf{R}, \tag{5.21}$$

where the constant $b_{\mathbf{R}}$ only depends on which one of the two $\mathbb{S}_N$ irreps in (5.20) $B_{IJ}$ belongs to, but not on the specific component.

    We now further impose the $\mathbb{Z}_N$ symmetry. Since the $\mathbb{S}_N$ irrep $\mathbf{N-1}$ does not contain any $\mathbb{Z}_N$ invariant, all $\lfloor N/2 \rfloor$ $\mathbb{Z}_N$ invariant $B$ deformations belong to the $\mathbb{S}_N$ irrep $\frac{(\mathbf{N-1})(\mathbf{N-2})}{\mathbf{2}} = \Lambda^2(\mathbf{N-1})$. Let us determine $b_{\frac{(\mathbf{N-1})(\mathbf{N-2})}{2}}$ below.[17]

    As discussed in Section 3.2.1, the $\mathbb{Z}_N$ invariant $B$-deformations are

$$B_a = B_{1,a+1} = B_{2,a+2} = \cdots = B_{N-a,N} = B_{N-a+1,1} = \cdots B_{N,a}, \tag{5.22}$$

for $a = 1, \cdots, N-1$. Only $\lfloor \frac{N}{2} \rfloor$ of them are independent since $B_a = -B_{N-a}$. Using (5.10) with $G_{IJ} = G_0$, we have

$$\frac{\mathrm{d}}{\mathrm{d}\log\mu}B_a = \frac{1}{2\pi G_0}\sum_{K\neq 1,1+a}(B_{1,1+a} - B_{1K} - B_{K,1+a}) = \frac{N}{2\pi}\frac{B_a}{G_0}. \tag{5.23}$$

That is, $b_{\frac{(\mathbf{N-1})(\mathbf{N-2})}{2}} = \frac{N}{2\pi}$.

    The physical coupling is not $B_a$ but $\widetilde{B}_a \equiv G_0^{-3/2}B_a$. To see this, we note that $B$ is the coefficient of a three-form flux $H$ on the target space. $H$ integrates to an order one number on the target space, i.e. $\int \mathrm{d}^3\phi H \sim 1$. Hence, the true coupling is $B$ divided by the cubic power of the radius (note that $G_0$ is proportional to the radius square on the target space). The flow equation for the physical coupling $\widetilde{B}_a = G_0^{-3/2}B_a$ is

$$\frac{\mathrm{d}}{\mathrm{d}\log\mu}\widetilde{B}_a = \frac{\widetilde{B}_a}{2\pi G_0}\left(\frac{N}{4} - \frac{3}{2}\right) + \cdots, \tag{5.24}$$

which is irrelevant for $N > 6$. The $\cdots$ represents higher-loop corrections to the beta function.[18]

## 6   $SU(3)/U(1)^2$

In this section we focus on the special case $SU(3)/U(1)^2$ sigma model, which was studied extensively in [1, 7].

---

[17]More precisely, the $\theta_I$'s transform in $\mathbf{N-1}$, which does not have a $\mathbb{Z}_N$ invariant. Yet, because of their $2\pi$-periodicity, they do have $\mathbb{Z}_N$ invariant values, which we use heavily. However, since $\theta_I$ do not affect the perturbative behavior of our theory, they can be ignored in this discussion.

[18]We thank I. Affleck and M. Lajko for discussions on the beta function of $\widetilde{B}_a$.

## 6.1 The Lagrangian

We start with the most general Lagrangian (2.12) without $a_I$, but before eliminating $\phi_i^3$.

$$\sum_{I=1}^3 \left( r_I \left( \sum_i |\partial \phi_i^I|^2 - |\sum_i \bar\phi_I^i \partial \phi_i^I|^2 \right) + \frac{\theta_I}{2\pi} \epsilon^{\mu\nu} \sum_i \partial_\mu \phi_i^I \partial_\nu \bar\phi_I^i \right)$$
$$+ \sum_{1 \le I < J \le 3} (G_{IJ} \delta^{\mu\nu} + B_{IJ} \epsilon^{\mu\nu}) \left( \sum_i \phi_i^I \partial_\mu \bar\phi_J^i \right) \left( \sum_j \bar\phi_I^j \partial_\nu \phi_j^J \right). \tag{6.1}$$

We can rewrite the Lagrangian by substituting

$$\phi_i^3 = e^{i\alpha} \Phi_i$$
$$\Phi_i \equiv \sum_{jk} \epsilon_{ijk} \bar\phi_1^j \bar\phi_2^k, \tag{6.2}$$

with some phase $\alpha$, which is related to $\det\phi = e^{i\alpha}$. With this we find (using (2.4))

$$\partial \alpha + \sum_{I=1}^3 a_I = 0, \tag{6.3}$$

and therefore we can choose the gauge $\alpha = 0$ and remain with $U(1)^2$ gauge freedom in the phases of $\phi_i^I$ with $I = 1, 2$. In the following we will replace $\phi_i^3$ by

$$\phi_i^3 = \sum_{jk} \epsilon_{ijk} \bar\phi_1^j \bar\phi_2^k. \tag{6.4}$$

We use

$$\sum_i |\partial \phi_i^3|^2 - |\sum_i \bar\phi_3^i \partial \phi_i^3|^2 = \sum_{I=1}^2 \left( \sum_i |\partial \bar\phi_I^i|^2 - |\sum_i \bar\phi_I^i \partial \phi_i^I|^2 \right) - 2 \sum_{jk} \bar\phi_2^k \phi_j^2 \partial \bar\phi_1^j \partial \phi_k^1$$

$$\epsilon^{\mu\nu} \sum_i \partial_\mu \phi_i^3 \partial_\nu \bar\phi_3^i = -\epsilon^{\mu\nu} \sum_{I=1}^2 \sum_i \partial_\mu \phi_i^I \partial_\nu \bar\phi_I^i$$

$$\left( \sum_i \phi_i^3 \partial_\mu \bar\phi_I^i \right) \left( \sum_j \bar\phi_3^j \partial_\nu \phi_j^I \right) = \sum_i \partial_\mu \bar\phi_I^i \partial_\nu \phi_i^I - \sum_{J=1}^2 \sum_{ij} \phi_i^J \partial_\mu \bar\phi_I^i \bar\phi_J^j \partial_\nu \phi_j^I \tag{6.5}$$

to write (6.1) as

$$\sum_{I=1}^2 \left( (r_I + r_3 + G_{I3}) \left( \sum_i |\partial \phi_i^I|^2 - |\sum_i \bar\phi_I^i \partial \phi_i^I|^2 \right) + \frac{\theta_I - \theta_3 + 2\pi B_{I3}}{2\pi} \epsilon^{\mu\nu} \sum_i \partial_\mu \phi_i^I \partial_\nu \bar\phi_I^i \right)$$
$$+ ((G_{12} - G_{13} - G_{23} - 2r_3) \delta^{\mu\nu} + (B_{12} - B_{13} + B_{23}) \epsilon^{\mu\nu}) \left( \sum_i \phi_i^1 \partial_\mu \bar\phi_2^i \right) \left( \sum_j \bar\phi_1^j \partial_\nu \phi_j^2 \right). \tag{6.6}$$

This indeed shows that some of the coefficients in (6.1) are redundant. The Lagrangian does not change under $r_I \to r_I + c_I$ combined with $G_{IJ} \to G_{IJ} - c_I - c_J$, so we can set $G_{IJ} = 0$. Similarly, the Lagrangian does not change under $\theta_I \to \theta_I + 2\pi b_I$ combined with $B_{IJ} \to B_{IJ} - b_I + b_J$, so we can set some of them to zero, e.g. $\theta_3 = B_{I3} = 0$. We conclude that we can write (6.1) as

$$\sum_{I=1}^2 (r_I + r_3) \left( \sum_i |\partial \phi_i^I|^2 - |\sum_i \bar\phi_I^i \partial \phi_i^I|^2 \right) + \sum_{I=1}^2 \frac{\theta_I}{2\pi} \epsilon^{\mu\nu} \sum_i \partial_\mu \phi_i^I \partial_\nu \bar\phi_I^i$$
$$- (2r_3 \delta^{\mu\nu} - B_{12} \epsilon^{\mu\nu}) \left( \sum_i \phi_i^1 \partial_\mu \bar\phi_2^i \right) \left( \sum_j \bar\phi_1^j \partial_\nu \phi_j^2 \right). \tag{6.7}$$

However, in this form some of the global symmetries at special values of the parameters are not manifest.

If we impose the $\mathbb{Z}_3$ symmetry, the parameters are restricted to $r_I = 0$, $G_{12} = G_{23} = G_{31} = G_0$, $B_{12} = B_{23} = B_{31} = B$, and $\theta_I = 2\pi n I/3$ as in (3.4). Using the redundancy $r_I \to r_I + c_I$ combined with $G_{IJ} \to G_{IJ} - c_I - c_J$, we can alternatively set $G_{IJ} = 0$ and $r_1 = r_2 = r_3 = G_0/2$ at the $\mathbb{Z}_3$ symmetric point. The $\mathbb{Z}_3$ symmetric Lagrangian is parameterized by $G_0$, $B$, and $n = 0, 1, 2$:

$$\sum_{I=1}^{3} \left( \frac{G_0}{2} \left( \sum_i |\partial \phi_i^I|^2 - |\sum_i \bar{\phi}_i^i \partial \phi_i^I|^2 \right) + \frac{\theta_I}{2\pi} \epsilon^{\mu\nu} \sum_i \partial_\mu \phi_i^I \partial_\nu \bar{\phi}_I^i \right)$$
$$+ B \sum_{I=1}^{3} \epsilon^{\mu\nu} \left( \sum_i \phi_i^I \partial_\mu \bar{\phi}_{I+1}^i \right) \left( \sum_j \bar{\phi}_I^j \partial_\nu \phi_j^{I+1} \right), \quad \text{with } \theta_I = \frac{2\pi I n}{3}. \tag{6.8}$$

Our parameters $G_0$ and $B$ are related to the $g$ and $\lambda$ in [1] as $G_0 = 1/g$ and $B = \lambda/(2\pi)$.

## 6.2 Renormalization Group Flow

### 6.2.1 $\mathbb{Z}_3$ Invariant Flow

Let us impose the $\mathbb{Z}_3$ symmetry and set $G_0 = G_{12} = G_{23} = G_{13}$ and $B = B_{12} = B_{23} = B_{31}$. From the one-loop beta functions (5.9) and (5.10), we obtain

$$\frac{d}{d\log\mu} G_0 = \frac{5}{4\pi} + \cdots, \tag{6.9}$$

$$\frac{d}{d\log\mu} B = \frac{3}{2\pi} \frac{B}{G_0} + \cdots, \tag{6.10}$$

and the flow equation for the physical coupling $\tilde{B} = G_0^{-\frac{3}{2}} B$ is

$$\frac{d}{d\log\mu} \widetilde{B} = -\frac{3}{8\pi} \frac{\widetilde{B}}{G_0}, \tag{6.11}$$

which means that $\widetilde{B}$ is relevant at one-loop.

### 6.2.2 Away from the $\mathbb{Z}_3$ Symmetric Point

Here we discuss the renormalization group flows of $G_{IJ}$ near the $\mathbb{Z}_3$ symmetric point $G_0 \equiv G_{12} = G_{23} = G_{13}$. In fact at this point the one-loop beta function of $G_{IJ}$ even enjoys a bigger $\mathbb{S}_3$ symmetry that will be useful below.

The one-loop beta functions (5.9) for $G_{IJ}$ in the $N = 3$ case are:

$$\frac{d}{d\log\mu} G_{12} = \frac{1}{2\pi} \left[ 3 + \frac{1}{2} \left( \frac{G_{12}^2}{G_{13} G_{23}} - \frac{G_{13}}{G_{23}} - \frac{G_{23}}{G_{13}} \right) \right] + \mathcal{O}(1/G_{IJ}),$$
$$\frac{d}{d\log\mu} G_{23} = \frac{1}{2\pi} \left[ 3 + \frac{1}{2} \left( \frac{G_{23}^2}{G_{13} G_{12}} - \frac{G_{12}}{G_{13}} - \frac{G_{13}}{G_{23}} \right) \right] + \mathcal{O}(1/G_{IJ}), \tag{6.12}$$
$$\frac{d}{d\log\mu} G_{13} = \frac{1}{2\pi} \left[ 3 + \frac{1}{2} \left( \frac{G_{13}^2}{G_{23} G_{12}} - \frac{G_{12}}{G_{23}} - \frac{G_{23}}{G_{23}} \right) \right] + \mathcal{O}(1/G_{IJ}).$$

Let us move away from $\mathbb{S}_3$ symmetric point by expanding $G_{IJ}$ as

$$\begin{aligned} G_{12} &= G_0 + \delta G_1, \\ G_{23} &= G_0 + \delta G_2, \\ G_{13} &= G_0 - \delta G_1 - \delta G_2, \end{aligned} \tag{6.13}$$

with $\delta G_a \ll G_0$. Note that these deformations not only break the $\mathbb{S}_3$ symmetry, but also the $\mathbb{Z}_3$ symmetry. Under $\mathbb{S}_3$, $G_0$ is in the trivial representation $\mathbf{1}$ while $\delta G_1, \delta G_2$ are in the doublet $\mathbf{2}$. The one-loop renormalization group flows to leading order in $\mathcal{O}(\delta G_a/G_0)$ are

$$\frac{\mathrm{d}}{\mathrm{d}\log\mu}G_0 = \frac{5}{4\pi} + \mathcal{O}\left(\frac{1}{G_0}, \left(\frac{\delta G_a}{G_0}\right)^2\right), \tag{6.14}$$

$$\frac{\mathrm{d}}{\mathrm{d}\log\mu}\delta G_a = \frac{3}{4\pi}\frac{\delta G_a}{G_0} + \mathcal{O}\left(\frac{1}{G_0}, \left(\frac{\delta G_a}{G_0}\right)^2\right), \quad a = 1, 2, \tag{6.15}$$

where the correction terms come from both the higher loop contributions suppressed by powers of $1/G_0$ as well as higher order terms in the small deviation $\delta G_a/G_0$. Let us understand the above beta functions using the $\mathbb{S}_3$ symmetry. First note that $\mathbf{2} \otimes \mathbf{2} = \mathbf{1} \oplus \mathbf{1}' \oplus \mathbf{2}$ where $\mathbf{1}'$ is the singlet representation that is odd under the odd permutations. Since $G_0$ is in the $\mathbf{1}$ of $\mathbb{S}_3$, the $\delta G_a/G_0$ correction can only arise at quadratic order to make a $\mathbf{1}$ out of two $\mathbf{2}$'s. On the other hand, since $\delta G_a$ is in the $\mathbf{2}$, its beta function to leading order must be proportional to itself.

The beta function of $\widetilde{\delta G_a}$ is

$$\frac{\mathrm{d}}{\mathrm{d}\log\mu}\widetilde{\delta G_a} = -\frac{1}{2\pi}\frac{1}{G_0}\widetilde{\delta G_a} + \cdots. \tag{6.16}$$

This means that the relative metric $\widetilde{\delta G_a}$ deviated from the $\mathbb{Z}_3$ symmetric point is relevant under the flow.

This behavior is consistent with what we saw earlier about the $\mathbb{Z}_N$ violating deformations of the IR WZW model.

# 7 General Flag Manifolds

## 7.1 General Considerations

Following the strategy in the previous sections, below we give a general discussion on flag manifold sigma models, both encompassing the well-known $\mathbb{CP}^{N-1}$ model and extending the analysis to more general flag manifolds. Consider the flag manifold

$$\mathcal{M}_{N_1, N_2, \cdots, N_m} = \frac{U(N)}{U(N_1) \times U(N_2) \times \cdots U(N_m)}, \qquad \sum_{I=1}^{m} N_I = N. \tag{7.1}$$

The case $N_I = 1$ for all $I$ reduces to the flag manifold that was discussed in the previous sections. The case $N_1 = 1$ and $N_2 = N - 1$ is the $\mathbb{CP}^{N-1}$ model.

The sigma model on (7.1) can be constructed in a similar way as in Section 2.1 using $N \times N$ complex scalar fields $\phi_i^I$ subject to the unitarity constraint (2.1) and the $U(N_1) \times U(N_2) \times \cdots \times U(N_m)$ gauge symmetry. The model has $m-1$ $\theta$-angles, and also various $r_I, G_{IJ}, B_{IJ}$ parameters as in (2.12). In the case of $SU(N)/U(1)^{N-1}$, each $r_I$, $G_{IJ}$, and $B_{IJ}$ term is individually gauge invariant. For the more general flag manifold (7.1), only certain linear combinations of them are gauge invariant.

The continuous global symmetry of the model is $PSU(N)$, which acts on the $i$ index of $\phi_i^I$. When the $\theta$ angles are either 0 or $\pi$, there is discrete symmetries including the $\mathbb{Z}_2$ global symmetry that maps $\phi_i^I \to \bar{\phi}_I^i$ and $a_I \to -a_I^*$. If some of the $N_I$'s are identical, we might also have an enhanced discrete symmetry for some special choices of the parameters. For example, in the case of $SU(N)/U(1)^{N-1}$, we have a $\mathbb{Z}_N$ global symmetry that cyclically permutes the

$U(1)$'s when the parameters are chosen at the $\mathbb{Z}_N$ symmetric configuration as discussed in Section 3.2.

After giving a basic description of the sigma model, we then ask whether for special choice of the parameters, this sigma model can flow to a gapless phase. In two-dimensional unitary compact CFT, continuous global symmetry always enhances to a full-fledged current algebra. Hence the most natural candidate for the gapless phase is the $SU(N)_k$ WZW model.

We review our strategy to analyze this putative flow from the sigma model to the WZW model:

1. First, we look for a deformation of the UV WZW model by certain potentials to restrict the fundamental field $U$ to take values in the flag $\mathcal{M}_{N_1,N_2,\cdots,N_m}$.[19] This restriction gives an embedding of the global symmetry $G_{UV}$ of the sigma model into the WZW symmetry, i.e. $G_{UV} \subset G_{WZW}$. Furthermore, the anomaly, if any, must match in the two theories. This procedure ensures that the flow is kinematically allowed.

2. Next, we ask the dynamical question. Using the embedding $G_{UV} \subset G_{WZW}$, we ask whether the WZW CFT has any $G_{UV}$-preserving relevant deformation that would take us away from the fixed point. If yes, the flow would miss the WZW fixed point without any fine-tuning. If no, there is a range of parameters the flow from the sigma model can hit the fixed point.

Let us start with step 1. The more general flag manifold sigma model $\mathcal{M}_{N_1,N_2,\cdots,N_m}$, with special choices of the parameters, can be embedded into the $SU(N)_k$ WZW model as

$$
\begin{aligned}
U &= \phi\Omega_0\phi^\dagger, \\
\Omega_0 &= \mathrm{diag}(\underbrace{e^{i\alpha_1},\cdots,e^{i\alpha_1}}_{N_1},\cdots,\underbrace{e^{i\alpha_m},\cdots,e^{i\alpha_m}}_{N_m}), \\
&\quad \sum_{I=1}^m N_I\alpha_I = 0 \mod 2\pi, \quad \alpha_I \neq \alpha_J \text{ if } I \neq J,
\end{aligned}
\tag{7.2}
$$

where $U \in SU(N)$ is the fundamental field of the WZW model. The constraint on the sum of $\alpha_i$'s is such that $\det U = 1$. The angles $\alpha_i$'s have to be distinct so that $\phi$ and $\phi V$ are identified only when $V \in U(N_1)\times\cdots\times U(N_m)$. By substituting (7.2) into the WZW action, we obtain the action (4.13), (4.18), and (4.20) in terms of the $\phi$ field.

We then proceed to step 2. Dynamically, there are generally symmetry-preserving relevant operators in the $SU(N)_k$ WZW model that can be generated and take us away from the fixed point. In this case, the flow generically misses the WZW fixed point without fine-tuning. For example, the relevant deformations in the $SU(N)_1$ WZW model are tabulated in Table 1.

To forbid such a flow away from the fixed point, one would like to impose as much discrete symmetry as possible, under which the relevant deformations are charged. In the case of $SU(N)/U(1)^{N-1}$ sigma model at special values of the parameters, we have a $PSU(N)\times\mathbb{Z}_N$ symmetry that forbids the relevant deformations in the $SU(N)_1$ WZW model. However, as discussed in Section 4.3, for higher level $k > 1$, the symmetry $PSU(N)\times\mathbb{Z}_N$ is not enough to remove some of the relevant deformations in the $SU(N)_k$ WZW model.

Below we first apply this strategy to the familiar $\mathbb{CP}^{N-1}$ sigma models and derive some known results on the flows.

Next, we consider a special case of (7.1), but still more general than the flag manifold $SU(N)/U(1)^{N-1}$. Let $N = rM$ where $r, M \in \mathbb{N}$. Consider the flag manifold:

$$
\frac{U(rM)}{U(M)^r}.
\tag{7.3}
$$

---

[19]See footnote 12 for a general construction of the required potential.

The $r = 2$ case is a Grassmannian. The $r = N$ case is the flag manifold (1.2) studied in the previous sections. We will argue that for $N$ sufficiently large and $r > 2$, the sigma model hits the $SU(N)_1$ WZW CFT fixed point.

## 7.2 $\mathbb{CP}^{N-1}$

Let us illustrate the above idea in the classic example of the $\mathbb{CP}^{N-1}$ model. We will discuss how the global symmetry and anomalies of the $\mathbb{CP}^{N-1}$ model are embedded into the $SU(N)_1$ WZW model. In this subsection we focus on the case when $N > 2$, while the case of $\mathbb{CP}^1$ is more special and deserves a separate discussion in Section 7.3.

In the $\mathbb{CP}^{N-1}$ model, we have $N$ complex scalar fields $z_i$ satisfying

$$\sum_{i=1}^{N} |z_i|^2 = 1 \,. \tag{7.4}$$

There is a $U(1)$ gauge transformation acting on $z^I$ as

$$z_i \to e^{i\lambda} z_i \,. \tag{7.5}$$

There is also an $SU(N)$ symmetry that rotates the different $z_i$'s in the fundamental representation. The center $\mathbb{Z}_N$ coincides with the $U(1)$ gauge symmetry, and should therefore be excluded from the global symmetry. The continuous global symmetry of the $\mathbb{CP}^{N-1}$ model is hence $PSU(N)$.

Let us discuss the discrete symmetry. Consider the charge conjugation $\mathbb{Z}_2^{charge}$ that acts as[20]

$$\mathbb{Z}_2^{charge} : \quad z_i \to \bar{z}_i \,, \tag{7.6}$$

where bar denotes complex conjugation. $\mathbb{Z}_2^{charge}$ also flips the signs of the gauge field, and hence that of the $\theta$ angle as well. It follows the $\mathbb{CP}^{N-1}$ model has a $\mathbb{Z}_2^{charge}$ charge conjugation symmetry only when $\theta$ is either $0$ or $\pi$.

To summarize, at $\theta = 0$ or $\pi$, the global symmetry $G_{UV}$ of the $\mathbb{CP}^{N-1}$ with $N > 2$ model is

$$\mathbb{CP}^{N-1} : \quad G_{UV} = PSU(N) \rtimes \mathbb{Z}_2^{charge} \,, \quad N > 2 \,. \tag{7.7}$$

Let us study the mixed anomaly between $PSU(N)$ and the charge conjugation $\mathbb{Z}_2^{charge}$, following a similar discussion in [12]. At $\theta = 0$, there is no mixed anomaly. Let us restrict to $\theta = \pi$ from now on. As in Section 2.2, we turn on a $PSU(N)$ background. With this background, we can add to the Lagrangian a counterterm $\frac{2\pi}{N} p w_2(\mathcal{P})$, where $p$ is an integer mod $N$. Under the $\mathbb{Z}_2^{charge}$ charge conjugation

$$(\theta = \pi, p) \underset{\mathbb{Z}_2^{charge}}{\to} (\theta = -\pi, -p) \sim (\theta = \pi, -p + 1) \,. \tag{7.8}$$

We would like to impose $p = -p + 1$ mod $N$ for some choice of $p$. If this can be done, then there is no anomaly. Otherwise there is a mixed anomaly between $PSU(N)$ and $\mathbb{Z}_2^{charge}$.

When $N$ is odd, we can choose $p = (N + 1)/2$ such that the above configuration goes back to itself. Hence there is no mixed anomaly when $N$ is odd at $\theta = \pi$.

When $N$ is even, however, there does not exist a choice of the counterterm $p$ such that the configuration is $\mathbb{Z}_2^{charge}$ invariant. This means that there is a mixed anomaly between $PSU(N)$ and $\mathbb{Z}_2^{charge}$ at $\theta = \pi$ when $N$ is even.

---

[20]We use the same notation as the $\mathbb{Z}_2^{charge} : \phi_i^I \to \bar{\phi}_i^i$ symmetry (3.14) in the flag sigma model $SU(N)/U(1)^{N-1}$ because they reduce to the same symmetry in the case of $\mathbb{CP}^1$. We will come back to this in Section 7.3.

We now discuss how the global symmetry of the $\mathbb{CP}^{N-1}$ model can be embedded into the $SU(N)_1$ WZW model. To do this, we consider the following restriction of the WZW fundamental field:[21]

$$
\begin{aligned}
U &= \phi \Omega_0 \phi^\dagger, \\
\Omega_0 &= e^{i\alpha} \text{diag}(1, e^{i\beta}, e^{i\beta}, \cdots, e^{i\beta}),
\end{aligned}
\tag{7.9}
$$

where $\phi \in U(N)$. For $U$ to be an element of $SU(N)$, we need $N\alpha + (N-1)\beta = 0$ mod $2\pi$. There is a $U(1) \times U(N-1)$ gauge symmetry acting from the right of $\phi$ that leaves $U$ invariant. Therefore the distinct $\phi$'s take value in $\frac{U(N)}{U(1) \times U(N-1)} = \mathbb{CP}^{N-1}$. We can use the $U(N-1)$ gauge symmetry and the unitarity constraint to solve all the $\phi_i^I$'s in terms of $\phi_i^{I=1}$. The latter is identified as the $z_i$ coordinates discussed earlier:

$$
z_i = \phi_i^{I=1}.
\tag{7.10}
$$

We now substitute (7.9) into the WZ term as in Section 4.2.1 to determine the $\theta$-angle in the $\mathbb{CP}^{N-1}$ model

$$
\begin{aligned}
\frac{i}{12\pi} k \int_{M_3} \text{Tr}[(U^\dagger dU)^3] = {} & \frac{k\alpha}{2\pi} \int_{M_2} \epsilon^{\mu\nu} \sum_i \partial_\mu \bar{\phi}_1^i \partial_\nu \phi_i^1 + \frac{k(\beta+\alpha)}{2\pi} \sum_{A=2}^{N} \int_{M_2} \epsilon^{\mu\nu} \sum_i \partial_\mu \bar{\phi}_A^i \partial_\nu \phi_i^A \\
& - \frac{k}{2\pi} \sum_{A=2}^{N} \sin\beta \, \epsilon^{\mu\nu} \int_{M_2} \left( \sum_i \phi_i^1 \partial_\mu \bar{\phi}_A^i \right) \left( \sum_j \bar{\phi}_A^j \partial_\nu \phi_j^1 \right).
\end{aligned}
\tag{7.11}
$$

The Lagrangian takes the form of (2.12) with $B_{1A} = -\frac{k}{2\pi}\sin\beta$, $\theta_1 = k\alpha$, and $\theta_A = k(\beta+\alpha)$. Although the Lagrangian is the same as the flag manifold case with a special choice of the parameters, now the $\phi$ fields are subject to the $U(1) \times U(N-1)$ gauge symmetry instead. We have the same redundancies of the parameters $\theta_I \to \theta_I + 2\pi b_I$ combined with $B_{IJ} \to B_{IJ} - b_I + b_J$, which follow from the constraint that $\phi_i^I$ is unitary. Setting $b_1 = -\frac{k}{2\pi}\sin\beta$ and $b_A = 0$, we can eliminate the $B_{IJ}$ term. The $\theta$ angles for the $U(1)$ and $U(N-1)$ gauge fields are then $\theta_{U(1)} = k\alpha - k\sin\beta$ and $\theta_{U(N-1)} = k(\beta+\alpha)$. The physical $\theta$-angle of the $\mathbb{CP}^{N-1}$ model is the difference between the above two, which is

$$
\theta = k(\beta + \sin\beta).
\tag{7.12}
$$

Let us restrict to $\beta = \pi$ and $k = 1$ so that

$$
\Omega_0 = e^{i\alpha} \text{diag}(1, -1, -1, \cdots, -1).
\tag{7.13}
$$

This choice describes the deformation of the $SU(N)_1$ UV WZW model to the $\mathbb{CP}^{N-1}$ sigma model at $\theta = \pi$. The angle $\alpha$ is then restricted to obey

$$
\alpha = \frac{2\ell - (N-1)}{N} \pi,
\tag{7.14}
$$

for some integer $\ell$.

The charge conjugation $\mathbb{Z}_2^{charge}$ (7.6) in the $\mathbb{CP}^{N-1}$ model acts on the coordinates $\phi$ as $\phi_i^I \to \bar{\phi}_I^i$. It is embedded into the WZW model as

$$
\mathbb{Z}_2^{charge}: \quad \phi_i^I \to \bar{\phi}_I^i \quad \Rightarrow \quad U \to \phi^* \Omega_0 \phi^T = e^{2i\alpha} U^*.
\tag{7.15}
$$

---

[21] See footnote 12 for a general construction of the required potential.

Since we have embedded the global symmetry of the $\mathbb{CP}^{N-1}$ model at $\theta = \pi$ into the WZW model with a restriction of the field, the anomaly, if exists, in the former theory must be matched by that of the latter.

Let us take a closer look on this charge conjugation $\mathbb{Z}_2^{charge}$ when embedded into the WZW model. When $N$ is odd, we can choose $\alpha = 0$ to satisfy (7.14), so the charge conjugation in the $\mathbb{CP}^{N-1}$ model descends to $U \to U^*$ in the WZW model. When $N$ is even, we can choose $\alpha = \frac{\pi}{N}$. $\mathbb{Z}_2^{charge}$ descends to $U \to e^{2\pi i/N} U^*$ in the WZW model.[22]

In all cases above, the mixed anomaly (or the absence thereof) between the $PSU(N)$ and $\mathbb{Z}_2^{charge}$ in the $\mathbb{CP}^{N-1}$ model at $\theta = \pi$ are reproduced by the WZW model with the appropriately identified $\mathbb{Z}_2$. However, for $N > 2$ there are $G_{UV}$-preserving relevant deformations in the $SU(N)_1$ model that can take us away from the fixed point as can be seen from Table 1. Indeed, it is believed that for $N > 2$ the $\mathbb{CP}^{N-1}$ does not undergo a second order phase transition.

## 7.3 $\mathbb{CP}^1$

The $\mathbb{CP}^1$ sigma model is at the intersection of two series of models, $\mathbb{CP}^{N-1}$ and $SU(N)/U(1)^{N-1}$. It is special relative to both generalizations and deserves a separate treatment. We will discuss its global symmetry and anomalies using three different representations:

1. The $N = 2$ case of $\mathbb{CP}^{N-1}$ (the $z_i$ fields).

2. The $N = 2$ case of $SU(N)/U(1)^{N-1}$ (the $\phi_i^I$ fields).

3. Deformation of the $SU(2)_1$ WZW model (the $U$ field).

We first remind the reader about the relations between the above three presentations. The $\mathbb{CP}^{N-1}$ field $z_i$ is related to $\phi_i^I$ via (7.10). In the $N = 2$ case, this becomes

$$\phi = \begin{pmatrix} \phi_{i=1}^{I=1} & \phi_{i=1}^{I=2} \\ \phi_{i=2}^{I=1} & \phi_{i=2}^{I=2} \end{pmatrix} = \begin{pmatrix} z_1 & -\bar{z}_2 \\ z_2 & \bar{z}_1 \end{pmatrix}. \tag{7.16}$$

The WZW fundamental field $U$ is related to $\phi$ as (7.9)

$$U = \phi \begin{pmatrix} -i & 0 \\ 0 & i \end{pmatrix} \phi^\dagger. \tag{7.17}$$

The $PSU(2)$ continuous global symmetry acts on the three fields as, $\vec{z} \to V\vec{z}$, $\phi \to V\phi$, $U \to VUV^\dagger$, respectively.

There are various $\mathbb{Z}_2$ symmetries. In Section 3.2 we discuss a $\mathbb{Z}_{N=2}$ symmetry (3.3) in $SU(2)/U(1)$, which acts as

$$\mathbb{Z}_2: \begin{pmatrix} z_1 \\ z_2 \end{pmatrix} \to \begin{pmatrix} -\bar{z}_2 \\ \bar{z}_1 \end{pmatrix}, \quad \phi_i^I \to \phi_i^{I+1}, \quad U \to -U. \tag{7.18}$$

Also, we discuss a $\mathbb{Z}_2^C$ symmetry (3.12) in Section 3.3, which can be thought of as the charge conjugation in the flag sigma model. It acts as

$$\mathbb{Z}_2^C \in PSU(2): \begin{pmatrix} z_1 \\ z_2 \end{pmatrix} \to \begin{pmatrix} -z_2 \\ z_1 \end{pmatrix}, \quad \phi_i^I \to \bar{\phi}_{N-I+1}^i, \quad U \to U^*. \tag{7.19}$$

---

[22]In the special case when $N = 2 \mod 4$, we can actually choose $\alpha = \pi/2$ so that $\Omega_0 = (i, -i, -i \cdots, -i)$. In this case $\mathbb{Z}_2^{charge}$ in the $\mathbb{CP}^{N-1}$ model descends to $U \to -U^*$. Recall that the global symmetry of the WZW model contains a $\mathbb{Z}_N \rtimes \mathbb{Z}_2$. When $N$ is even, there is a $\mathbb{Z}_2$ subgroup of the $\mathbb{Z}_N$ that commutes with the other $\mathbb{Z}_2$. The charge conjugation $\mathbb{Z}_2^{charge}$ is the diagonal of the above two $\mathbb{Z}_2$'s.

$\mathbb{Z}_2^C$ is in fact an element of $PSU(2)$ by choosing $V = \begin{pmatrix} 0 & -1 \\ 1 & 0 \end{pmatrix}$. Finally, from the perspective of the $\mathbb{CP}^{N-1}$ model, we have $\mathbb{Z}_2^{charge}$ defined in (7.6), which acts on the three fields as

$$\mathbb{Z}_2^{charge} = \text{diag}(\mathbb{Z}_2^C \times \mathbb{Z}_2): \quad \begin{pmatrix} z_1 \\ z_2 \end{pmatrix} \to \begin{pmatrix} \bar{z}_1 \\ \bar{z}_2 \end{pmatrix}, \quad \phi_i^I \to \bar{\phi}_I^i, \quad U \to -U^*. \tag{7.20}$$

Note that this is also the $\mathbb{Z}_2^{charge}$ in (3.14) we defined for the flag sigma model $SU(N)/U(1)^{N-1}$. It is the composition of (7.19) and (7.18).

Each $\mathbb{Z}_2$ above is natural from at least one perspective, but might seem ad hoc from another. The $\mathbb{Z}_2$ in (7.18) has the distinguished feature that it commutes with $PSU(2)$, while the other two don't since they involve a $PSU(2)$ element.

$\mathbb{CP}^1$ sigma model is a perfect example illustrating that there is generally no canonical choice of the charge conjugation symmetry. What might be called the charge conjugation in one presentation might not even involve any complex conjugation in another presentation. For example, while the $\mathbb{Z}_2^C$ action on the $\phi$ field involves complex conjugation, its action on $z_i$ does not.

To summarize the discussion so far, the $\mathbb{CP}^1$ sigma model at $\theta = 0, \pi$ has global symmetry

$$\mathbb{CP}^1: \quad G_{UV} = PSU(2) \times \mathbb{Z}_2 = O(3), \tag{7.21}$$

where the $\mathbb{Z}_2$ is given in (7.18). For $\theta = \pi$ it is embedded into the symmetry of the WZW model as in (4.3).

As analyzed in Section 3.2, there is a mixed anomaly between the two factors in (7.21) at $\theta = \pi$ [12, 23]. On the other hand, there is no mixed anomaly between $\mathbb{Z}_2^C$ and $PSU(2)$ because the former is an element of the latter and there is no pure anomaly of $PSU(2)$.

So far we have embedded the global symmetry $PSU(2) \times \mathbb{Z}_2$ of the $\mathbb{CP}^1$ sigma model at $\theta = \pi$ into the $SU(2)_1$ WZW model and match the anomaly. Dynamically, we need to ask if there is a symmetry-preserving relevant deformation in the WZW CFT that can take us away from the fixed point. Unlike the $N > 2$ case, the only $PSU(2)$-invariant relevant deformation $\mathcal{O}_1$ (in the notation of Section 4.3) in the $\mathbf{2}$ of $SU(2)$ is odd under the $\mathbb{Z}_2$ (7.18), and is therefore forbidden. Indeed, it is well-known that the $\mathbb{CP}^1$ sigma model at $\theta = \pi$ does flow to the $SU(2)_1$ WZW model [3, 16, 24].

## 7.4 $U(rM)/U(M)^r$

Let us move on to the more general flag manifold $\frac{U(N)}{U(M)^r}$ (7.3) with $N = rM$.[23] This sigma model, for special choice of the parameters, has a $\mathbb{Z}_r$ global symmetry. It is a straightforward generalization of the $\mathbb{Z}_N$ symmetry in Section 3.2 in the $SU(N)/U(1)^{N-1}$ sigma model.

The $\mathbb{Z}_r$ symmetry acts on the $\phi_i^I$ field as

$$\mathbb{Z}_r: \quad \phi_i^I \to \phi_i^{I+M}, \tag{7.22}$$

and cyclically permutes the $r$ $U(M)$ gauge fields.

We can embed this target space into the $SU(N)$ WZW fundamental field as[24]

$$U = \phi \Omega_0 \phi^\dagger,$$
$$\Omega_0 = e^{\frac{-2\pi i(r-1)}{2r}} \text{diag}(\underbrace{1, \cdots, 1}_{M}, \underbrace{e^{\frac{2\pi i}{r}}, \cdots, e^{\frac{2\pi i}{r}}}_{M}, \cdots, \underbrace{e^{\frac{2\pi i(r-1)}{r}}, \cdots, e^{\frac{2\pi i(r-1)}{r}}}_{M}). \tag{7.23}$$

---

[23]The case of $\mathbb{CP}^1$, which corresponds to $N = 2$, $M = 1$, $r = 2$ is as always special and has already been considered separately in Section 7.3. Below we will exclude that case.

[24]See footnote 12 for the potential that enforces this field restriction.

Then the $\mathbb{Z}_r$ symmetry acts on the WZW fundamental field as

$$\mathbb{Z}_r : \quad U \to e^{\frac{2\pi i}{r}} U . \tag{7.24}$$

There are $r$ $\theta$-angles associated to the $r$ $U(M)$ gauge fields. From (7.23), we see that they take the following $\mathbb{Z}_r$ symmetric values, $\theta_a = \frac{2\pi a}{r}$, $a = 1, 2, \cdots, r$.

There is also a charge conjugation symmetry (3.12)

$$\mathbb{Z}_2^C : \quad \phi_i^I \to \bar{\phi}_{N-I+1}^i , \qquad U \to U^* , \tag{7.25}$$

where we have used $(\Omega_0)_I^* = (\Omega_0)_{N-I+1}$.

The global symmetry of the sigma model (7.3) is then

$$G_{UV} = (PSU(N) \times \mathbb{Z}_r) \rtimes \mathbb{Z}_2^C . \tag{7.26}$$

$G_{UV}$ is embedded into the WZW symmetry $G_{WZW} = \frac{SU(N)_L \times SU(N)_R}{\mathbb{Z}_N} \rtimes \mathbb{Z}_2^C$ via (7.23).

Are there $G_{UV}$-preserving relevant deformations in the $SU(N)_1$ WZW model that can take us away from the fixed point? Using Table 1, we see that when $N \geq 10$ and $r \geq 3$, there is no $G_{UV}$-preserving relevant deformation, and we expect the sigma model to hit the $SU(N)_1$ WZW CFT fixed point.[25] The other possibility is $r = N$ and $M = 1$, which is the flag manifold $SU(N)/U(1)^{N-1}$ that we have already discussed. As before, we do not expect it to hit the higher level WZW model without fine-tuning, because of the relevant deformation caused by the primary in the adjoint representation.

## 8 Summary of Results

We have studied various aspects of the (1+1)-dimensional sigma model on the flag manifold $U(N)/U(M)^r$ with $N = rM$. Imposing the $\mathbb{Z}_r$ global symmetry, we argue that if

- $r = N$ and $M = 1$, i.e. the flag manifold $SU(N)/U(1)^{N-1}$, or

- $r \geq 3$ and $N \geq 10$,

the $U(N)/U(M)^r$ sigma model with the $r$ $U(M)$ $\theta$-angles chosen to be $\theta_a = n\frac{2\pi a}{r}$ (with $gcd(n, r) = 1$) flows to a gapless phase described by the $SU(N)_1$ WZW model. This generalizes the flow from the $\mathbb{CP}^1$ sigma model at $\theta = \pi$ to the $SU(2)_1$ WZW model and the flag with $N = 3$ of [1]. (For larger values of $N$ see also the discussion in [4], which slightly differs from ours.) Our argument is based on the following facts:

- Kinematics: The global symmetries and their anomalies in the flag sigma model can be embedded into the WZW model.

- Dynamics: The fixed point is robust in the sense that there is no symmetry-preserving relevant deformation that can potentially take us away from the WZW CFT.

As we emphasized in the introduction, these arguments make it possible to find the $SU(N)_1$ theory at long distances, but they do not guarantee it.

---

[25]Because of the marginal deformations in the $SU(8)_1$ and the $SU(9)_1$ WZW CFT discussed in footnote 14, a more detailed analysis is needed for the flag models $U(8)/U(4)^2$ and $U(9)/U(3)^3$.

## Acknowledgements

We are particularly grateful to I. Affleck, who suggested we study this sigma model and for numerous useful comments. Input from C. Cordova, S. Komatsu, M. Lajko, H.T. Lam, Y.-H. Lin, R. Mahajan, K. Wamer, and E. Witten is also acknowledged. The work of K.O. and S.H.S. is supported in part by the National Science Foundation grant PHY-1606531 and that of N.S. by DOE grant DE-SC0009988. K.O. is also supported by the Paul Dirac fund. S.H.S. is also supported by the Roger Dashen Membership.

## A The Trace Conditions

In this appendix we show that if a unitary matrix $U$ satisfies

$$\text{Tr}[U^n] = 0\,, \quad n = 1, 2, \cdots, \lfloor N/2 \rfloor\,, \tag{A.1}$$

then

$$\text{Tr}[U^n] = 0\,, \quad n = 1, 2, \cdots, N-1\,. \tag{A.2}$$

Let us start with a formula relating the determinant of a matrix to its traces:

$$\det(U) = (-1)^N \sum_{k_1, k_2, \cdots, k_N} \prod_{n=1}^N \frac{(-1)^{k_n}}{n^{k_n} k_n!} \text{Tr}[U^n]^{k_n}\,, \tag{A.3}$$

where the sum is over all non-negative integers $k_n$ such that

$$\sum_{n=1}^N n k_n = N\,. \tag{A.4}$$

Among all the tuples $(k_1, k_2, \cdots, k_N)$, every term except for $(0, 0, \cdots, 1)$ has at least one $k_n$ nonzero with $n \leq \lfloor N/2 \rfloor$. It follows that if (A.1) is satisfied, then

$$\det(U) = (-1)^N \frac{1}{N} \text{Tr}[U^N]\,. \tag{A.5}$$

Taking the absolute value on both sides, we have

$$|\text{Tr}[U^N]| = N\,. \tag{A.6}$$

Since the eigenvalues of a unitary matrix are all phases, the above equation is only possible if all eigenvalues of $U^N$ are the same, i.e.

$$U^N = e^{i\alpha} I\,. \tag{A.7}$$

Using the complex conjugate of (A.1):

$$0 = \text{Tr}[(U^\dagger)^n] = e^{-i\alpha} \text{Tr}[U^{N-n}]\,, \quad n = 1, 2, \cdots, \lfloor N/2 \rfloor\,. \tag{A.8}$$

This completes the proof.

# B  Counting Invariant Deformations

## B.1  Generality

In this appendix we will enumerate $PSU(N)$ and $PSU(N) \times \mathbb{Z}_N$ invariant deformations in the $SU(N)/U(1)^{N-1}$ sigma model by embedding the latter into the WZW model.

Let us consider the possible $PSU(N)$ invariant deformations in terms of $U$ satisfying (4.7). In particular, (4.7) (see also (4.10)) implies that

$$U^N = (-1)^{N-1} I. \tag{B.1}$$

This gives the following identity that will be useful later:

$$0 = d(U^N) = dU U^{N-1} + U dU U^{N-2} + \cdots + U^{N-1} dU. \tag{B.2}$$

Also, (B.1) implies that we can always replace $U^\dagger$ by powers of $U$:

$$U^\dagger = (-1)^{N-1} U^{N-1}. \tag{B.3}$$

Demanding $PSU(N)$ invariance, there are two types of two-derivative terms we can write down. We will call them the type $G$ and type $B$ terms for reasons that will become obvious momentarily. They are

$$G: \quad \text{Tr}[U^a dU \wedge \star U^b dU], \quad a \geq b \geq 0, \tag{B.4}$$

$$B: \quad \text{Tr}[U^a dU \wedge U^b dU], \quad a > b \geq 0. \tag{B.5}$$

Note that the type $B$ term vanishes if $a = b$. Thus without loss of generality, we will assume $a > b$ for such a term.

Let us now rewrite the above deformations in terms of $\phi$'s. The relation between the two bases is $U = \phi \Omega_0 \phi^\dagger$. A general $G$ type term $\text{Tr}[U^a dU \wedge \star U^b dU]$ can be expanded as follows

$$\text{Tr}[U^a dU \wedge \star U^b dU] = \text{Tr}[\phi \Omega_0^a \phi^\dagger (d\phi \Omega_0 \phi^\dagger + \phi \Omega_0 d\phi^\dagger) \wedge \star \phi \Omega_0^b \phi^\dagger (d\phi \Omega_0 \phi^\dagger + \phi \Omega_0 d\phi^\dagger)]$$
$$= \text{Tr}[2\Omega_0^{a+1} \phi^\dagger d\phi \wedge \star \Omega_0^{b+1} \phi^\dagger d\phi - \Omega_0^a \phi^\dagger d\phi \wedge \star \Omega_0^{b+2} \phi^\dagger d\phi - \Omega_0^{a+2} \phi^\dagger d\phi \wedge \star \Omega_0^b \phi^\dagger d\phi],$$

where we have used $d\phi \phi^\dagger + \phi d\phi^\dagger = 0$. All three terms above are of the following form, which can be computed straightforwardly:

$$\text{Tr}[\Omega_0^A \phi^\dagger d\phi \Omega_0^B \phi^\dagger \wedge \star d\phi] = \omega^{-\frac{N-1}{2}(A+B)} \sum_{i,j,I,J} \omega^{A(I-1)+B(J-1)} (\bar{\phi}_I^i \partial^\mu \phi_i^J)(\bar{\phi}_J^j \partial_\mu \phi_j^I)$$
$$= -\omega^{-\frac{N+1}{2}(A+B)} \sum_{I,J} \omega^{AI+BJ} (\sum_i \bar{\phi}_I^i \partial^\mu \phi_i^J)(\sum_j \phi_j^I \partial_\mu \bar{\phi}_J^j). \tag{B.6}$$

Summing over three such terms, we obtain the $G$ type term in the $\phi$ language ($a \leq b$, $a, b = 0, 1, \cdots, N-2$)

$$\text{Tr}[U^a dU \wedge \star U^b dU]$$
$$= \omega^{-\frac{N+1}{2}(a+b+2)} \sum_{1 \leq I < J \leq N} (\omega^{aI+bJ} + \omega^{aJ+bI})(\omega^I - \omega^J)^2 \delta^{\mu\nu} (\sum_i \bar{\phi}_I^i \partial_\mu \phi_i^J)(\sum_j \phi_j^I \partial_\nu \bar{\phi}_J^j). \tag{B.7}$$

Similarly, the $B$ type term can be written in the $\phi$ basis as ($a < b$, $a, b = 0, 1, \cdots, N-1$)

$$\text{Tr}[U^a dU \wedge U^b dU]$$
$$= \omega^{-\frac{N+1}{2}(a+b+2)} \sum_{1 \leq I < J \leq N} (\omega^{aI+bJ} - \omega^{aJ+bI})(\omega^I - \omega^J)^2 \epsilon^{\mu\nu} (\sum_i \bar{\phi}_I^i \partial_\mu \phi_i^J)(\sum_j \phi_j^I \partial_\nu \bar{\phi}_J^j). \tag{B.8}$$

## B.2 $PSU(N)$ Invariant Deformations

In this subsection we will count the number of $PSU(N)$ invariant deformations from the WZW model point of view. Naively, the range of $a, b$ is $0, 1, 2, \cdots, N-1$. However, because of (B.2), there are relations between terms with different $a, b$. If $a = N-1$, we can rewrite such a type $G$ term as

$$
\begin{aligned}
\mathrm{Tr}[U^{N-1}\mathrm{d}U \wedge \star U^b\mathrm{d}U] = &-\mathrm{Tr}[U^{N-2}\mathrm{d}U \wedge \star U^{b+1}\mathrm{d}U] - \mathrm{Tr}[U^{N-3}\mathrm{d}U \wedge \star U^{b+2}\mathrm{d}U] - \cdots \\
&-\mathrm{Tr}[U^b\mathrm{d}U \wedge \star U^{N-1}\mathrm{d}U] - \mathrm{Tr}[U^{b+1}]\mathrm{d}U \wedge \star U^N\mathrm{d}U] - \cdots - \mathrm{Tr}[\mathrm{d}U \wedge \star U^{b+N-1}\mathrm{d}U].
\end{aligned}
\tag{B.9}
$$

Since the $\wedge\star$ product is symmetric, we can move the first term in the second line to the left, and solve for $\mathrm{Tr}[U^{N-1}\mathrm{d}U \wedge \star U^b\mathrm{d}U]$ in terms of other type $G$ terms with $a < N-1$. We conclude that it suffices to restrict the range of $a, b$ to $0, 1, \cdots, N-2$ for the type $G$ terms.

However, the same argument does not hold for the type $B$ terms, because the first term in the second line above now becomes $\mathrm{Tr}[U^{N-1}\mathrm{d}U \wedge U^b\mathrm{d}U]$, which is the same as the LHS. In fact, this identity is trivial for the type $B$ terms and cannot help us solving terms with $a = N-1$ in terms of others.

To conclude, we record the ranges of $a, b$ below for the type $G$ and $B$ terms:

$$
G: \ \mathrm{Tr}[U^a\mathrm{d}U \wedge \star U^b\mathrm{d}U], \quad a \geq b, \ a, b = 0, 1, 2, \cdots, N-2,
\tag{B.10}
$$

$$
B: \ \mathrm{Tr}[U^a\mathrm{d}U \wedge U^b\mathrm{d}U], \quad a > b, \ a, b = 0, 1, 2, \cdots, N-1.
\tag{B.11}
$$

Let us now count the $PSU(N)$ invariant deformations. For the type $G$ terms, there are

$$
\binom{N-1}{2} + (N-1) = \frac{N(N-1)}{2}
\tag{B.12}
$$

terms. These are the $G$-deformation in the $\phi_i^I$ language. On the other hand, there are

$$
\binom{N}{2} = \frac{N(N-1)}{2}
\tag{B.13}
$$

type $B$ terms, which are the $B$-deformation in the $\phi_i^I$ language. These altogether reproduce the counting of $G$ and $B$ deformations in Section 2.1 in the $\phi$ language.

## B.3 $PSU(N) \times \mathbb{Z}_N$ Invariant Deformations

Next, we further impose the $\mathbb{Z}_N$ invariance condition. Recall that $\mathbb{Z}_N$ acts on $U$ as

$$
\mathbb{Z}_N: \ U \to \omega U.
\tag{B.14}
$$

Hence the $\mathbb{Z}_N$ invariance condition further demands the following relation between $a$ and $b$ for both type $G$ and $B$ terms:

$$
\mathbb{Z}_N \text{ invariance}: \ a + b + 2 = 0 \bmod N.
\tag{B.15}
$$

Let us enumerate the number of $PSU(N) \times \mathbb{Z}_N$ invariant deformations. When $N$ is even, we have the following choices of $(a, b)$:

$$
G: \ (N-2, 0), \ (N-3, 1), \ \cdots, \left(\frac{N-2}{2}, \frac{N-2}{2}\right),
\tag{B.16}
$$

$$
B: \ (N-2, 0), \ (N-3, 1), \ \cdots, \left(\frac{N-2}{2}+1, \frac{N-2}{2}-1\right).
\tag{B.17}
$$

Hence there are $N/2$ type $G$ terms and $(N-2)/2$ type $B$ terms when $N$ is even.

Next, when $N$ is odd, we have the following possible $(a, b)$:

$$G : (N-2, 0), (N-3, 1), \cdots, \left(\frac{N-1}{2}, \frac{N-3}{2}\right), \tag{B.18}$$

$$B : (N-2, 0), (N-3, 1), \cdots, \left(\frac{N-1}{2}, \frac{N-3}{2}\right). \tag{B.19}$$

Hence there are $(N-1)/2$ type $G$ terms and $(N-1)/2$ type $B$ terms when $N$ is odd.

To conclude, we have $\lfloor N/2 \rfloor$ $PSU(N) \times \mathbb{Z}_N$ invariant type $G$ terms, and $\lfloor (N-1)/2 \rfloor$ $PSU(N) \times \mathbb{Z}_N$ invariant type $B$ terms. This matches perfectly with the counting in the $\phi_i^I$ language. These altogether reproduce the counting of $\mathbb{Z}_N$ invariant $G$ and $B$ deformations in Section 3.2.1 in the $\phi$ language.

# C  Symmetric Space $SU(N)/SO(N)$

In the main text we discuss how the sigma models with target space (7.1) can be embedded into the $SU(N)_k$ WZW model by restricting the WZW fundamental field. This procedure automatically shows that the sigma model has the same anomaly as the $SU(N)_k$ WZW model, and can be applied to any submanifold of $SU(N)$. As another example, here we apply this strategy to the symmetric space $SU(N)/SO(N)$. This model is interesting because an integrable flow from the $SU(N)/SO(N)$ sigma model with nontrivial $\theta$ angle to the $SU(N)_1$ WZW fixed point is worked out in [25], generalizing the result for the $\mathbb{CP}^1 = SU(2)/SO(2)$ sigma model [26, 27].[26]

The symmetric space $SU(N)/SO(N)$ can be embedded into $SU(N)$ as

$$SU(N)/SO(N) \simeq \{U \in SU(N) | \exists \phi \in SU(N) \text{ s.t. } U = \phi\phi^T\}, \tag{C.1}$$

since a matrix $U = \phi\phi^T$ is invariant under the right action of an orthogonal matrix on $\phi$. This variable $\phi$ can be thought as the field of the $SU(N)/SO(N)$ sigma model. Furthermore, the same embedding can also be characterized by

$$SU(N)/SO(N) \simeq \{U \in SU(N) | U = U^T\}. \tag{C.2}$$

Obviously, a matrix $U = \phi\phi^T$ is symmetric. Conversely, a symmetric special unitary matrix $U$ can be represented as $U = \phi\phi^T$ with

$$\phi = OD^{\frac{1}{2}}O^T, \tag{C.3}$$

where $O$ is the orthogonal matrix diagonalizing $U$ as $OUO^T = D$, and $D^{\frac{1}{2}}$ is a matrix satisfying $(D^{\frac{1}{2}})^2 = D$ and $\det(D^{\frac{1}{2}}) = 1$.[27]

The isometry group $G_{\text{iso}}$ of $SU(N)/SO(N)$ is the subgroup of $\frac{SU(N)_L \times SU(N)_R}{\mathbb{Z}_N} \rtimes \mathbb{Z}_2^C$ (see Section 4.1) preserving the condition $U = U^T$. This means a pair $(V_L, V_R)$ inside the identity component of $G_{\text{iso}}$ should satisfies

$$V_L U V_R^\dagger = (V_L U V_R^\dagger)^T = V_R^* U V_L^T \tag{C.4}$$

for any $U = U^T$. This condition is equivalent to

$$V_R^* = \omega^{-\ell} V_L, \tag{C.5}$$

---

[26]We thank Ho Tat Lam for a discussion about this coset.

[27]Existence of such $O$ can be proven as follows: A symmetric unitary matrix $U$ can be written as $X + iY$ with real symmetric matrices $X$ and $Y$. Since $U$ is unitary, we have $\mathbb{1} = X^2 + Y^2 + i(XY - YX)$ and therefore $X$ and $Y$ commutes with each other. Hence we can take a orthogonal matrix $O$ which simultaneously diagonalizes $X$ and $Y$.

with some $\ell \in \mathbb{Z}_N$. Therefore, we can label an element of the identity component of $G_{\text{iso}}$ as $(V_L, \ell)$ with $V_L \in SU(N)$ and $\ell \in \mathbb{Z}_N$. The quotient in $(SU(N)_L \times SU(N)_R)/\mathbb{Z}_N$ implies

$$(V_L, V_R = \omega^\ell V_L^*) \sim (\omega V_L, \omega^{\ell+2}(\omega V_L)^*), \tag{C.6}$$

which, in terms of the labeling $(V_L, \ell)$, means

$$(V_L, \ell) \sim (\omega V_L, \ell + 2). \tag{C.7}$$

We can describe the group $G_{\text{iso}}$ as

$$G_{\text{iso}} = \frac{SU(N) \times \mathbb{Z}_N}{\mathbb{Z}_N} \rtimes \mathbb{Z}_2^C. \tag{C.8}$$

The action of the charge conjugation is $(V_L, \ell) \mapsto (V_L^*, -\ell)$. We can also write the same isometry group as

$$G_{\text{iso}} \simeq \frac{\widetilde{SU(N)}}{\mathbb{Z}_2} \rtimes \mathbb{Z}_2^C, \tag{C.9}$$

where

$$\widetilde{SU(N)} = \{\hat{V} \in U(N) | \det\hat{V} = \pm 1\}, \tag{C.10}$$

and $\mathbb{Z}_2$ quotient is generated by $-\mathbb{1} \in U(N)$. The isomorphism (C.9) is given by

$$(V_L, \ell) \mapsto \omega^{-\ell/2} V_L. \tag{C.11}$$

Note that the ambiguity of the definition of $\omega^{-\ell/2}$ is absorbed by the $\mathbb{Z}_2$ quotient of $\frac{\widetilde{SU(N)}}{\mathbb{Z}_2}$.

The restriction of the field $U$ in the WZW model to be symmetric can be realized by adding the potential

$$g \sum \text{Tr}((U - U^T)(U - U^T)^\dagger) = g \sum_{i,j} \left| U_{ij} - U_{ji} \right|^2 = g\text{Tr}(2\mathbb{1} - UU^* - (UU^*)^\dagger), \tag{C.12}$$

and then taking the coefficient $g$ to infinity.[28] This potential is invariant under the symmetry $G_{\text{iso}}$, since an element $(V_L, V_R)$ in $G_{\text{iso}}$ acts on the term $\text{Tr}UU^*$ transforms as

$$\text{Tr}(UU^*) \to \text{Tr}(V_L U V_R^\dagger V_L^* U^* V_R^T) = \text{Tr}(V_L U(\omega^\ell V_L^T) V_L^* U^*(\omega^{-\ell} V_L^\dagger)) = \text{Tr}(UU^*), \tag{C.13}$$

because of (C.5). This shows that this potential indeed restricts the field to the desired subspace.

# D  Supersymmetric Flag Sigma Model

In this appendix we consider the $\mathcal{N} = (2, 2)$ sigma model on $SU(N)/U(1)^{N-1}$.

## D.1  Kähler Moduli Space

For concreteness, we will restrict to the $SU(3)/U(1)^2$ sigma model in this subsection.

$\mathcal{N} = (2, 2)$ supersymmetry requires the target space to be a Kähler manifold. However, not all the $SU(3)$ invariant metrics on the flag manifold are Kähler. By the corollary in IV.5 of [28], the dimension of the $SU(3)$ invariant Kähler moduli space is 2. We would like to determine how this subspace of Kähler metric is embedded into the larger metric moduli space.

---

[28]We can alternatively use the more general construction described in footnote 12.

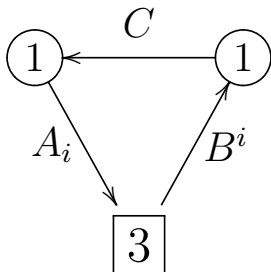

Figure 2: The gauged linear sigma model description of the flag manifold $SU(3)/U(1)^2$. Each node represents a $U(1)$ vector multiplet and the square represents the $SU(3)$ flavor symmetry. $A_i, B^i$ and $C$ denote the chiral superfields in the fundamental representation of the two gauge groups connected by the corresponding arrow. $i = 1, 2, 3$ is the flavor $SU(3)$ index. There is a superpotential term $W = \sum_{i=1}^{3} A_i B^i C$.

To determine the Kähler moduli space, it will be useful to have a gauged linear sigma model description of the model. The $SU(3)$ model can be described by a gauged linear sigma model via the Abelian quiver shown in Figure 2 with a superpotential $W = \sum_{i=1}^{3} A_i B^i C$.

Let us analyze the global symmetry of this quiver theory. The superpotential $W = \sum_{i=1}^{3} A_i B^i C$ breaks the two $SU(3)$'s that rotate $A_i$ and $B^i$ independently to the diagonal $SU(3)$. Furthermore, the $\mathbb{Z}_3$ center of the diagonal $SU(3)$ acts in the same way on $A_i$ and $B^i$ as the two $U(1)$ gauge symmetries, and should therefore be excluded from the global symmetry. To conclude, the continuous global symmetry of the quiver is $PSU(3)$, consistent with our description of the model in terms of $\phi$ in Section 2.1.

The space of zeros of the classical potential of this quiver gauge theory is as follows.[29] It is described by the complex coordinates $(A_i, B^i, C)$ satisfying the D-term equations:

$$-|C|^2 + \sum_{i=1}^{3} |A_i|^2 = \zeta_1, \quad -|C|^2 + \sum_{i=1}^{3} |B^i|^2 = \zeta_2, \quad \text{(D.1)}$$

($\zeta_I$ are the coefficients of Fayet-Iliopoulos (FI) terms) and the F-term equations:

$$\sum_{i=1}^{3} A_i B^i = 0, \quad CA_i = 0, \quad CB^i = 0, \quad \text{(D.2)}$$

and it should be subject to the identification under the $U(1)^2$ gauge symmetry. Here $\zeta_I$ are the two Kähler moduli of the quiver moduli space.

We will take the FI parameters $\zeta_{1,2}$ to be both positive. For this choice of signs, the F-term equation sets

$$C = 0. \quad \text{(D.3)}$$

The quiver moduli space is now described by the complex coordinates $(A_i, B^i)$ satisfying

$$\sum_{i=1}^{3} |A_i|^2 = \zeta_1, \quad \sum_{i=1}^{3} |B^i|^2 = \zeta_2, \quad \sum_{i=1}^{3} A_i B^i = 0, \quad \text{(D.4)}$$

---

[29]In similar higher dimensional systems this is the moduli space of classical vacua.

and modulo the $U(1)^2$ gauge transformation:

$$U(1)_1 : \quad A_i \sim e^{i\theta_1} A_i, \quad B^i \to B^i,$$
$$U(1)_2 : \quad A_i \sim A_i, \quad B^i \to e^{i\theta_2} B^i. \tag{D.5}$$

We recognize that the variables $A_i$ and $B^i$ are related to the $\phi$ fields in (6.7) as

$$A_i = \sqrt{\zeta_1}\, \phi_i^{I=1}, \qquad B^i = \sqrt{\zeta_2}\, \bar{\phi}_{I=2}^i. \tag{D.6}$$

Note that $\phi_i^{I=3}$ has already been solved in terms of $\phi_i^{I=1,2}$ (to be more precise, their complex conjugates) via (6.4). Also note that the choice of the complex structure in the $\phi$ language is different than that in the quiver presentation.

The two Kähler parameters $\zeta_{1,2}$ parameterize a real two-dimensional subspace $\mathcal{M}_K$ of the total three-dimensional moduli space $\mathcal{M}$ of the $SU(3)$ invariant metrics on $SU(3)/U(1)^2$.

Let us write down the bosonic part of the Lagrangian for the gauged linear sigma model. In the IR of the model, both the gauge coupling and the superpotential coupling flow to infinity so that $A_i, B^i$ are restricted to obey (D.4) with $C = 0$. In this limit we can integrate out the two $U(1)^2$ gauge fields and write the Lagrangian solely in terms of $A_i, B^i$. This computation is identical to that in Section 2.1. We obtain

$$\mathcal{L}_{GLSM} = \sum_{I=1}^{2} \zeta_I \left( \sum_{i=1}^{3} |\partial \phi_i^I|^2 - |\sum_{i=1}^{3} \bar{\phi}_I^i \partial \phi_i^I|^2 \right) + \sum_{I=1}^{2} \frac{\theta_I}{2\pi} \epsilon^{\mu\nu} \sum_{i=1}^{3} \partial_\mu \phi_i^I \partial_\nu \bar{\phi}_I^i. \tag{D.7}$$

Since we start with the gauged linear sigma model description, this Lagrangian depends only on the 2 Kähler moduli (along with the corresponding $\theta$ angles) but not the most general $SU(3)$ invariant metric deformations. Comparing $\mathcal{L}_{GLSM}$ with (6.7), we find the embedding of the 2-dimensional subspace $\mathcal{M}_K$ into $\mathcal{M}$:

$$\mathcal{M}_K = \left\{ (r_1, r_2, r_3) \in \mathcal{M} \,\middle|\, r_3 = 0 \right\}. \tag{D.8}$$

Let us discuss the discrete symmetry action on the moduli space. The $\mathbb{S}_3$ symmetry acts on the moduli space $\mathcal{M}$ by permuting the $r_I$'s. There is a $\mathbb{Z}_2$ subgroup, which exchanges $r_1$ with $r_2$, of $\mathbb{S}_3$ that leaves $\mathcal{M}_K$ invariant. The Kähler moduli $\zeta_{1,2}$ fall into the $\mathbf{1}$ and $\mathbf{1}'$ representations of the $\mathbb{Z}_2$, which are paired with the two $\theta$ angles in the same $\mathbb{Z}_2$ representations.

The locus $r_I = 0$ on $\mathcal{M}$ naively gives a divergent Ricci tensor by inspecting (6.12). However, (6.12) is only valid when *all* the $r_I$'s are large and cannot be trusted on the submanifold $\mathcal{M}_K$.

## D.2  Twisted Chiral Ring

In this subsection we discuss the twisted chiral ring of the $SU(N)/U(1)^{N-1}$ sigma model with $\mathcal{N} = (2,2)$ supersymmetry. The twisted chiral ring of this model is worked out in [29]. Define the following $N \times N$ matrix

$$A = \begin{pmatrix} x_1 & q_1 & 0 & \dots & 0 \\ -1 & x_2 & q_2 & \dots & 0 \\ 0 & -1 & x_3 & \dots & 0 \\ \vdots & \vdots & \vdots & \ddots & q_{N-1} \\ 0 & 0 & 0 & -1 & x_N \end{pmatrix}. \tag{D.9}$$

In other words,

$$A_{ij} = x_i \delta_{ij} - 1\delta_{i,j+1} + q_i \delta_{i,j-1}. \tag{D.10}$$

The $q_i$'s are related to the $N-1$ complexified FI parameters. The twisted chiral ring is generated by $x_1, x_2, \cdots, x_N$ with relations given by the coefficients of $\lambda$ in the polynomial

$$-\lambda^N + \det(A + \lambda I). \tag{D.11}$$

For example, when $N = 2$, the relations are

$$x_1 + x_2 = 0, \quad x_1 x_2 + q_1 = 0, \tag{D.12}$$

which gives the familiar twisted chiral ring $\mathbb{C}[x_1]/\{(x_1)^2 = q_1\}$ of $\mathbb{CP}^1$.

Next, the relations in the $N = 3$ case are

$$\begin{aligned}
x_1 + x_2 + x_3 &= 0, \\
x_1 x_2 + x_2 x_3 + x_1 x_3 + q_1 + q_2 &= 0, \\
x_1 x_2 x_3 + q_2 x_1 + q_1 x_3 &= 0.
\end{aligned} \tag{D.13}$$

Using the first relation, we can solve $x_2$ as $-x_1 - x_3$ and get $x_1 x_3 (x_1 + x_3) = q_2 x_1 + q_1 x_3$ and $x_1^2 + x_1 x_3 + x_3^2 = q_1 + q_2$. Since both $q_1, q_2$ are nonzero, the relations forbid either $x_1$ or $x_3$ to be zero. Therefore, we can further simplify the two relations by multiplying the second relation by $x_1$ and $x_3$ to get

$$x_1^3 = q_1(x_1 - x_3), \quad x_3^3 = q_2(x_3 - x_1). \tag{D.14}$$

To conclude, the twisted chiral ring of $SU(3)/U(1)^2$ is

$$\mathbb{C}[x_1, x_3]/\{x_1^3 = q_1(x_1 - x_3), \, x_3^3 = q_2(x_3 - x_1)\}. \tag{D.15}$$

Let us reproduce this twisted chiral ring from the Abelian quiver description in Figure 2. Denote the bottom component scalar fields of the two $U(1)$ vector multiplets by $\Sigma_1$ and $\Sigma_2$, respectively. Integrating out the chiral multiplets $A_i, B^i, C$ gives us the following quantum twisted superpotential

$$\begin{aligned}
\tilde{W} =& it_1 \Sigma_1 + it_2 \Sigma_2 + \frac{3}{2\pi} \Sigma_1 (\log \Sigma_1 - 1) \\
& - \frac{3}{2\pi} \Sigma_2 (\log \Sigma_2 - 1) + \frac{1}{2\pi} (\Sigma_2 - \Sigma_1)(\log(\Sigma_2 - \Sigma_1) - 1),
\end{aligned} \tag{D.16}$$

where $t_i = ir_i + \theta_i/2\pi$ is the complexified FI parameters with periodicity $t_i \sim t_i + 1$. From this twisted superpotential, we obtain the following relations

$$\begin{aligned}
\Sigma_1^3 &= -e^{-2\pi i t_1}(\Sigma_1 - \Sigma_2), \\
\Sigma_2^3 &= e^{2\pi i t_2}(\Sigma_2 - \Sigma_1),
\end{aligned} \tag{D.17}$$

which agrees with (D.14) if we identify the variables on both sides as

$$x_1 = \Sigma_1, \quad x_3 = \Sigma_2, \quad q_1 = -e^{-2\pi i t_1}, \quad q_2 = e^{2\pi i t_2}. \tag{D.18}$$

There is an alternative non-Abelian quiver description (see Figure 3) of the $SU(3)/U(1)^2$ sigma model [30], which is related to the Abelian quiver in Figure 2 by a duality move [31]. We will derive the same twisted chiral ring from this non-Abelian quiver. The general equivalence of the twisted chiral rings from dual quivers is discussed in [32].

Let the bottom component of the $U(1)$ vector multiplet be $\Sigma'$, and let $\Sigma'_1, \Sigma'_2$ be those of the Cartan of the $U(2)$ vector multiplet. Let $t'_1$ and $t'_2$ be the complexified FI parameters of the $U(1)$ and the $U(2)$ nodes, respectively. The twisted superpotential is

$$\tilde{W} = it'_1 \Sigma' + it'_2(\Sigma'_1 + \Sigma'_2) - \frac{1}{2\pi} \sum_{a=1}^{2}(\Sigma'_a - \Sigma')\big(\log(\Sigma'_a - \Sigma') - 1\big) + \frac{3}{2\pi} \sum_{a=1}^{2} \Sigma'_a(\log \Sigma'_a - 1), \tag{D.19}$$

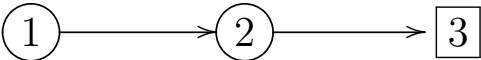

Figure 3: The non-Abelian gauged linear sigma model description of the flag manifold $SU(3)/U(1)^2$. It is related to the Abelian quiver in Figure 2 by a duality move on the middle node.

from which we obtain the following equations

$$
\begin{aligned}
e^{-2\pi i t_1'} &= (\Sigma_1' - \Sigma')(\Sigma_2' - \Sigma'), \\
e^{2\pi i t_2'} \Sigma_1'^3 &= \Sigma_1' - \Sigma', \\
e^{2\pi i t_2'} \Sigma_2'^3 &= \Sigma_2' - \Sigma'.
\end{aligned}
\tag{D.20}
$$

Note that $\Sigma_a'$'s are not good coordinates of the ring. The two fields $\Sigma_1'$ and $\Sigma_2'$ are exchanged by the $\mathbb{Z}_2$ Weyl group. The Weyl invariant coordinates are

$$
A = \Sigma_1' + \Sigma_2', \quad B = \Sigma_1'\Sigma_2'.
\tag{D.21}
$$

We can rewrite the above equations in terms of $\Sigma', A, B$:

$$
e^{-2\pi i t_1'} = B - A\Sigma' + \Sigma'^2,
\tag{D.22}
$$

$$
e^{2\pi i t_2'} A(A^2 - 3B) = A - 2\Sigma',
\tag{D.23}
$$

$$
A^2 - B = e^{-2\pi i t_2'}.
\tag{D.24}
$$

Importantly, we have assumed $\Sigma_1' \neq \Sigma_2'$ to avoid the massless W-bosons to obtain the above relations [33, 34]. We can use (D.24) to solve $B$ as $B = A^2 - e^{-2\pi i t_2'}$, and then substitute it back into (D.24) to obtain

$$
A^3 = e^{-2\pi i t_2'}(A + \Sigma').
\tag{D.25}
$$

Finally, we multiply (D.23) by $A + \Sigma'$ and use the last equation (D.25) to get

$$
\Sigma'^3 = e^{-2\pi i t_1'}(A + \Sigma').
\tag{D.26}
$$

We therefore have derived the same twisted chiral ring as (D.15) if we identify the variables as

$$
\Sigma_1 = \Sigma', \quad \Sigma_2 = -A,
\tag{D.27}
$$

$$
t_1 = t_1' + \frac{1}{2}, \quad t_2 = -t_2'.
\tag{D.28}
$$

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
