# Peer review of "Sigma Models on Flags"

_SciPost Physics, doi:SciPost Phys. 6, 017 (2019)_

## Round 2 · Referee Report · Paul Fendley (Referee 1) · 2018-12-7

Strengths

1) This paper discusses in great depth and clarity sigma models on flag manifolds $SU(N)/U(1)^{N-1}$, including topological ``theta’’ angles, whose applications are famous both in particle and condensed-matter theory.

2) The most exciting result of the analysis are the compelling arguments that there exist regions of couplings and theta angles where the flow is to a critical theory, the $SU(N)_1$ Wess-Zumino-Witten conformal field theory. This generalises the famous Haldane/Affleck result for N=2 at theta=pi and recent work by Lajko et al arXiv:1706.06598 for the N=3 case.

3) Since there are lattice models realising these field theories, this result greatly generalises the Haldane ``conjecture’’.

Weaknesses

Aren't really any

Report

It’s hard to imagine getting any further using this type of direct analysis than the authors already did. Thus I believe it should be published in SciPost.

Given that this forum is public, it seems appropriate to share a few thoughts on extending these results. The Haldane conjecture has been so well established by now it shouldn’t be called a conjecture, but these new results still are conjectures. So one very interesting direction to purse further would be to provide further evidence for them. My bias would be toward finding integrable models exhibiting these flows, the same way the Zamolodchikovs did for the flow in $CP^1$ at $\theta=\pi$ to $SU(2)_1$ (i.e. the Haldane case). Moreover, this would be interesting in its own right: I don’t know any integrable sigma models not on symmetric spaces or deformations of them (like the sausage model), at least for compact manifolds.

One strategy in finding such integrable models would be to exploit symmetries and (putative) integrability to guess the corresponding scattering matrix in the Zamoldchikov style. I showed long ago that this strategy works for the $SU(N)/SO(N)$ models at $\theta=\pi$ discussed in an appendix, and so might bear fruit in the flag-manifold case. A complementary strategy is to explore the connection with the homogenous sine-Gordon models analysed in a series of papers 15-20 years ago, culminating in a paper by Dorey and Miramontes in arXiv:hep-th/0405275. These models are integrable deformations of coset CFTs $G_k/U(1)^{rank(G)}$, known as Gepner parafermions. It has long been known due to work by Fateev and Al Zamolodchikov, generalised in an old paper arXiv:hep-th/9906036 of mine, that taking the $k,k’\to\infty$ limit of certain deformations of $G_k/H_{k’}$ coset models gives the sigma model $G/H$. Perhaps the same construction will work here.

Requested changes

1) The strategy the authors use (analyse symmetries of the action, then see if these are enough to forbid all relevant perturbations for the putative IR fixed point) in this context is mainly due to Ian Affleck. One particular paper of his most closely related to the strategy here, in particular the role of the $\mathbb{Z}_n$ in $SU(N)_1$, is Nucl. Phys. B305 (1988) 582. Thus a reference in the introduction to this paper (in addition to his others already there) would probably be a good idea. His Les Houches lectures from that year also provide a superb introduction to the topic.

2) in several places $PSU(N)=PSU(N)/\mathbb{Z}_n$ is written. Presumably what is meant is $PSU(N)=SU(N)/\mathbb{Z}_n$ ??

3) Since the bulk of the paper is on $SU(N)/U(1)^{N-1}$ (and then $U(N)/U(M)^r$), probably this should be mentioned in the abstraction in addition to the general case.

---

## Round 2 · Referee Report · Sunil Mukhi (Referee 2) · 2019-1-9

Strengths

  1. Studies a novel family of 2d QFT’s that generalise the well-known and interesting $CP^1$ sigma model. However they are a different class of generalisations from the more obvious $CP^N$ sigma models, so $CP^1$ lies at the intersection of these two families (as well as other families like $O(N)$ with very different behaviour)
  2. The new family studied here has multiple parameters, unlike $CP^N$ where there is a single parameter, the size of the manifold plus a theta angle. Thus richer physical effects can be identified.
  3. Restricting to what the authors call the “extreme” case and imposing a discrete $Z_N$ symmetry, one finds mixed anomalies indicative of a non-trivial IR fixed point, which is argued to be an $SU(N)_1$ WZW model.
  4. The authors set up a nice flow diagram between the UV of the WZW model, the UV of the flag manifold model and the IR theory of both cases (which is the WZW CFT). This illustrates in particular that the CFT can be reached without fine tuning if one is in the basin of attraction.

Weaknesses

No particular weaknesses

Report

An interesting paper that generalises some previous results on the IR limit of a family of bosonic nonlinear sigma models, presenting a carefully reasoned analysis of the possibilities. A supersymmetric generalisation is also discussed.

Requested changes

  1. On page 6, “flat manifold” should be “flag manifold”?
  2. This could just be my own confusion, and is anyway rather off the main point of the paper, but the last sentence in footnote 5 puzzled me. While considering the $S_N$ invariant theory and the possibility that its UV symmetry decouples in the IR, the authors acknowledge that it might, in principle, flow to a CFT without any relevant deformation - for example $E_{8,1}$. Then they invoke modular invariance to rule out this possibility for $c<8$ $(N <4)$. So far, so good. However then they go on to say “It is reasonable that the same conclusion is true for all $N$”. But why is that reasonable? The argument for $c<8$ certainly doesn't generalise to $c>8$, where we know many relatively simple CFT's that have no marginal deformations.

---

## Editorial Decision

published